# Fast, accurate, and interpretable decoding of electrocorticographic signals using dynamic mode decomposition
Ryohei Fukuma [1,2,8], Kei Majima[3,4,8], Yoshinobu Kawahara[5,6], Okito Yamashita[5,7], Yoshiyuki Shiraishi[1], Haruhiko Kishima [2] & Takufumi Yanagisawa [1,2] ✉

Dynamic mode (DM) decomposition decomposes spatiotemporal signals into basic oscillatory components (DMs). DMs can improve the accuracy of neural decoding when used with the nonlinear Grassmann kernel, compared to conventional power features. However, such kernel-based machine learning algorithms have three limitations: large computational time preventing real-time application, incompatibility with non-kernel algorithms, and low interpretability. Here, we propose a mapping function corresponding to the Grassmann kernel that explicitly transforms DMs into spatial DM (sDM) features, which can be used in any machine learning algorithm. Using electrocorticographic signals recorded during various movement and visual perception tasks, the sDM features were shown to improve the decoding accuracy and computational time compared to conventional methods. Furthermore, the components of the sDM features informative for decoding showed similar characteristics to the high-γ power of the signals, but with higher trial-to-trial reproducibility. The proposed sDM features enable fast, accurate, and interpretable neural decoding.

Fast and accurate characterization of the spatiotemporal dynamics of neural signals is crucial to decode neural signals for brain-computer interfaces (BCIs), which can be applied to allow severely paralyzed patients to reconstruct their lost communication and mobility functions. Dynamic mode decomposition (DMD) is a numerical method to obtain representations called Koopman modes[1–6], each of which corresponds to an oscillation of a spatial pattern with a fixed frequency and decay/growth rate. For a multi-dimensional ($P$-dimensional) time series $\mathbf{x}(t) \in \mathbb{R}^P$ that can be approximated as evolving over time $\Delta t$, as shown in Eq. (1), DMD approximately decomposes $\mathbf{x}(t)$ as a superposition of $K$ oscillatory components in a complex space, as shown in Eq. (2), by obtaining $\lambda_k$ and $\boldsymbol{\varphi}_k$ based on singular value decomposition (SVD) applied for $\mathbf{A}$ in Eq. (1) (see "Methods"):

$$\mathbf{x}(t + \Delta t) = \mathbf{A}\mathbf{x}(t) \tag{1}$$

$$\mathbf{x}(t) \approx \sum_{k=1}^{K} \boldsymbol{\varphi}_k \, r_k^{\,t} \exp(2\pi i \, f_k \, t) \, b_k \tag{2}$$

where $r_k = |\lambda_k|^{1/\Delta t}$ and $f_k = \arg(\lambda_k)/2\pi\Delta t$.

Here, each of the $K$ oscillatory components is represented by a spatial pattern $\boldsymbol{\varphi}_k$, an $P$-dimensional complex vector representing the dynamic mode (DM), and the following parameters of the $k$th DM: $f_k$, the frequency of the DM; $r_k$, the decay/growth rate of the DM; and $b_k$, a scalar that determines the initial phase of the DM.

Figure 1 shows examples of DMs. Spatiotemporal signals of $\mathbf{x}(t)$ were generated as the sum of two oscillations with different spatial distributions ($\mathbf{x}_1(t) + \mathbf{x}_2(t)$, Fig. 1a). When SVD was applied for $\mathbf{A}$ in Eq. (1), four singular values were obtained as nonzero values (Fig. 1b). By using the four SVD components corresponding to these nonzero singular values, four oscillatory components (DMD components) were acquired (Fig. 1c). By adding the products of each DM and time dynamics, an approximation for $\mathbf{x}(t)$ ($\mathbf{x}_{\text{recon}}(t)$) can be obtained (Fig. 1d). Notably, some of the DMD components have complex conjugate pairs for their modes and time dynamics (e.g., DMD components 1 and 2), yielding real summed values. Because the original observed spatiotemporal signals $\mathbf{x}(t)$ are strictly real, DMD always decomposes the signals into real DMD components or pairs of complex conjugate DMD components, ensuring that the reconstructed $\mathbf{x}(t)$ ($\mathbf{x}_{\text{recon}}(t)$) is in real space[7].

[1]Institute for Advanced Co-Creation Studies, Osaka University, Suita, Japan. [2]Department of Neurosurgery, Graduate School of Medicine, Osaka University, Suita, Japan. [3]Institute for Quantum Life Science, National Institutes for Quantum Science and Technology, Chiba, Japan. [4]JST PRESTO, Saitama, Japan. [5]RIKEN Center for Advanced Intelligence Project, Tokyo, Japan. [6]Graduate School of Information Science and Technology, Osaka University, Suita, Japan. [7]Department of Computational Brain Imaging, Neural Information Analysis Laboratories, ATR, Kyoto, Japan. [8]These authors contributed equally: Ryohei Fukuma, Kei Majima. ✉e-mail: tyanagisawa@nsurg.med.osaka-u.ac.jp

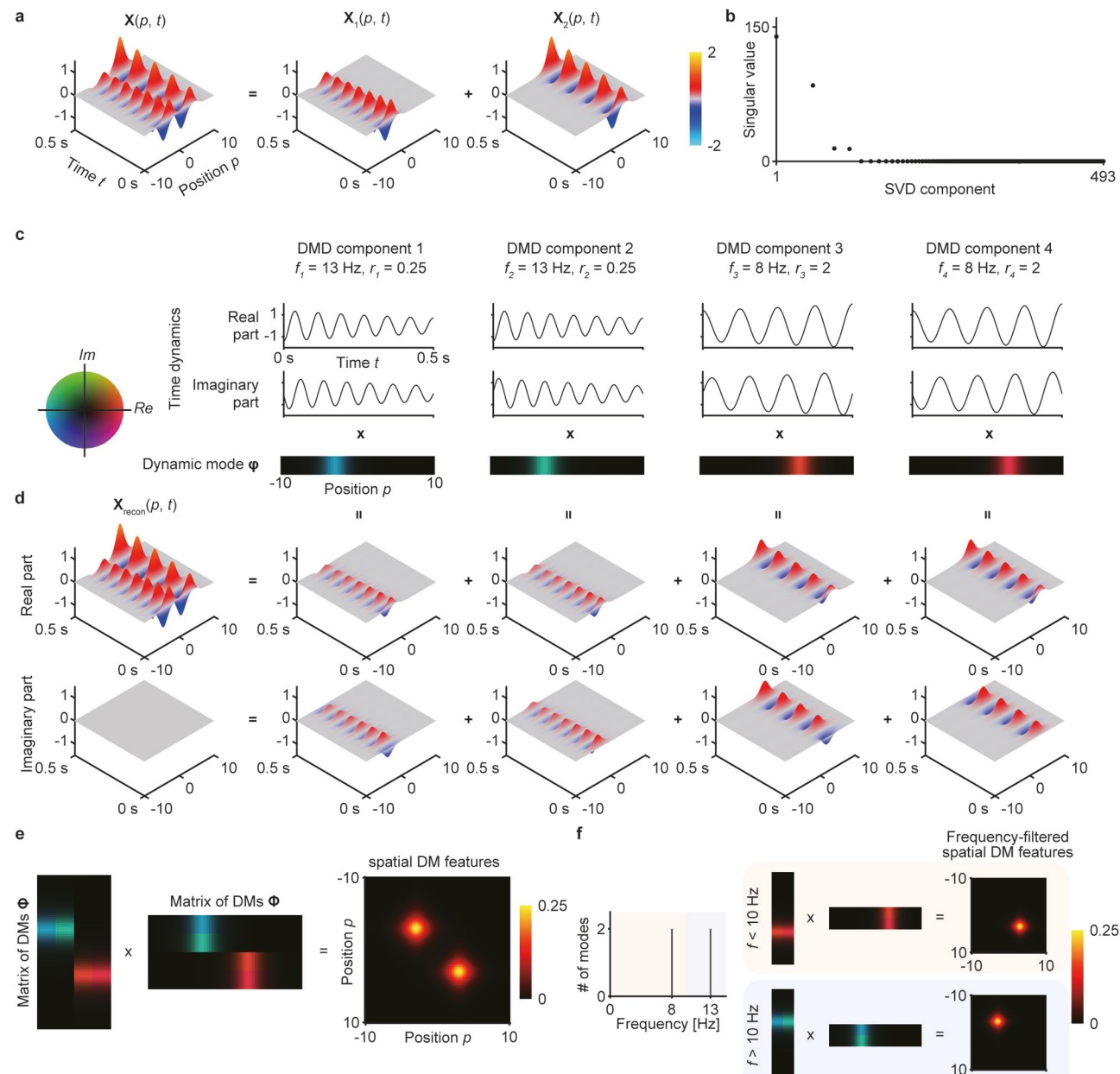

**Fig. 1 | Representative example of dynamic mode decomposition. a** Spatiotemporal signals of $\mathbf{x}(t)$ were created by adding two different signals: $\mathbf{x}_1(t)$: 13-Hz sine wave with decaying amplitude over time ($\mathbf{X}_1(p, t) = \text{sech}(p + 3) \times 0.25^t \times \sin(2\pi \times 13t)$); $\mathbf{x}_2(t)$: 8-Hz sine wave with increasing amplitude ($\mathbf{X}_2(p, t) = \text{sech}(p - 3) \times 2^t \times \sin(2\pi \times 8t)$). Both signals were sampled at 1 kHz for a duration of 0.5 s. Observation points (i.e., position $p$) ranged from $-10$ to 10, with an interval of 0.25. **b** Singular values acquired by SVD based on (**a**) are shown, starting with the largest value. All values except the first four SVD components were zero. Because of the stacking process prior to the SVD (see "Methods"), only 493 components were acquired by the SVD process. The horizontal axis is shown with a log scale. **c, d** Four DMD components determined based on the four SVD components with nonzero singular values are shown for (**c**) time dynamics and DMs, along with (**d**) their corresponding spatiotemporal signals. Here, DMD components 1 and 2 and DMD components 3 and 4 are complex conjugate pairs with respect to their modes and temporal dynamics because the original spatiotemporal signal $\mathbf{x}(t)$ is strictly real. For visibility, each DM was L2-normalized, and the scaling factor for each DM was applied to the corresponding time dynamics so that their products were the same. By adding the products of the four DMs and time dynamics in (**c**), the original signals ($\mathbf{x}(t)$) were reconstructed ($\mathbf{x}_{\text{recon}}(t)$) as shown in (**d**). **e** A matrix of the concatenated L2-normalized DMs was multiplied by its transpose to acquire the sDM features. **f** To acquire frequency-filtered sDM features, the L2-normalized DMs were filtered based on their corresponding frequencies before the multiplication was performed.

The Koopman modes extracted by the DMD process (DM, $\boldsymbol{\varphi}_k$) capture some characteristic spatiotemporal patterns in the dynamics of neurophysiological signals; thus, the DMs are useful for neural decoding. Previous studies have demonstrated that DMs characterize spindles recorded by electrocorticographic (ECoG) signals[8] and different traits of functional magnetic resonance imaging (fMRI) scans[9]. Our previous study also demonstrated that DMs were informative for classifying the ECoG signals corresponding to some hand movements[10]. In that study, DMD was applied

to the ECoG signals for each trial and the resultant matrix of DMs ($\boldsymbol{\Phi} = \left[\boldsymbol{\varphi}_1 \dots \boldsymbol{\varphi}_K\right]$) was compared with the matrix of another trial using the projection kernel[11–13], one of the Grassmann kernel functions, to quantify the similarity among all trials. It should be emphasized that a direct comparison between the matrix of DMs for the $i$th trial ($\boldsymbol{\Phi}^i = \left[\boldsymbol{\varphi}_1^i \dots \boldsymbol{\varphi}_K^i\right]$) and that of the $j$th trial $\boldsymbol{\Phi}^j$ is difficult, because each DM $\boldsymbol{\varphi}_k^i$ has a different frequency $f_k^i$. Hence, the projection kernel was introduced in the previous study to define the similarity between the matrices for each trial, $k_p(\boldsymbol{\Phi}^i, \boldsymbol{\Phi}^j)$.

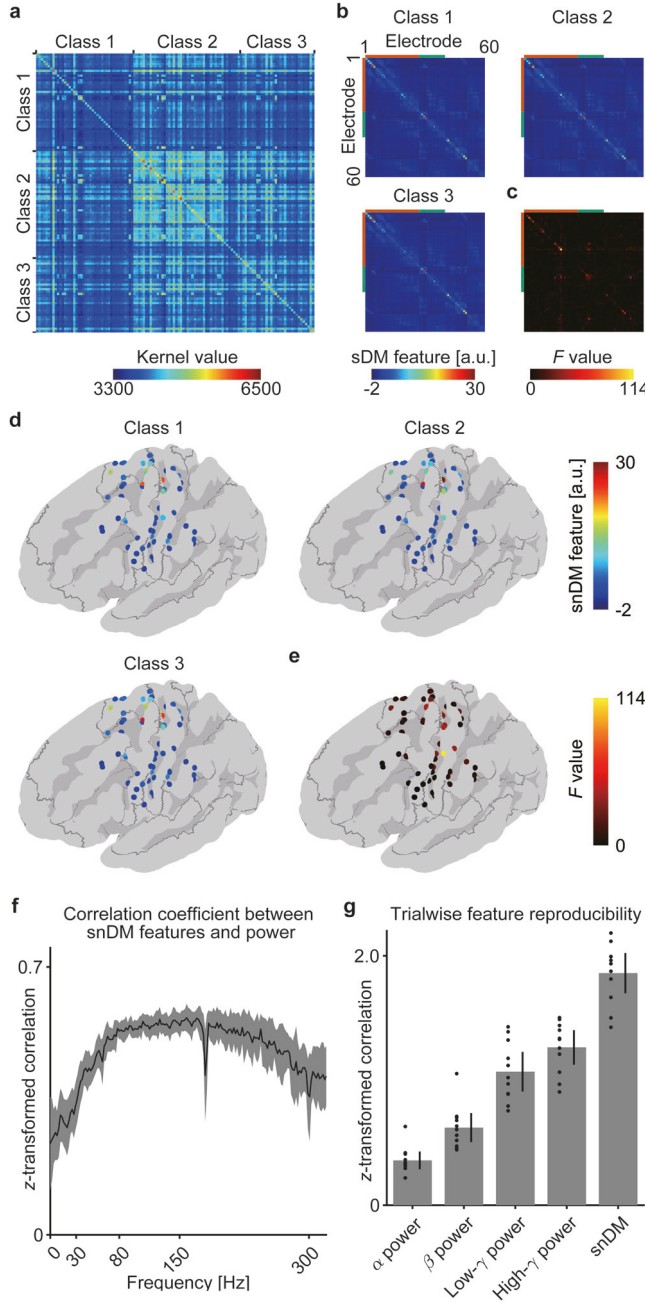

**Fig. 2 | Representative Gram matrix and sDM features during the arm motor task. a–e** The ECoG signals recorded when patient 1 performed three types of movements were used to calculate 300 DMs for each trial to obtain the Gram matrix acquired by projection kernel $k_p(\boldsymbol{\Phi}^i, \boldsymbol{\Phi}^j)$ among the trials. The sDM features were calculated based on the same DMs used in (**a**) to visualize the (**b**) averaged sDM features among the trials of the same movement type and (**c**) $F$ values of the sDM features among the different movement types. Notably, the sDM features are symmetrical due to their definition. The snDM features shown in (**b**) and their corresponding $F$ values (diagonal part of (**c**)) are shown by corresponding colors at the location of each electrode on the normalized brain in (**d**) and (**e**), respectively. **f** For each patient, Pearson's correlation coefficient was calculated between the snDM features and the PSD of each frequency by concatenating the features from all channels and all trials. The correlation coefficients were Fisher $z$-transformed and averaged among the patients, as shown by the black line, with the colored area representing the 95% confidence intervals (CIs) among the patients. The dips in correlation at 180 Hz and 300 Hz are considered to be due to noise caused by odd harmonics of the power supply frequency (60 Hz). **g** For the power features of each frequency band and the snDM features, the reproducibility of the features among the trials of the same movement type was evaluated with Pearson's correlation coefficients. To calculate the reproducibility of each feature, the correlation coefficients were calculated among all possible pairs of trials of the same movement type, Fisher $z$-transformed, and averaged for each patient. The average reproducibility is shown in the bar graph, with 95% CIs among the patients. Each dot represents the reproducibility of each patient.

time for predicting a new sample (trial) with a kernel-based machine learning model is proportional to the number of training samples; hence, it is difficult to obtain predictions in real time with a large training dataset. Furthermore, it is difficult to combine sparse regularization methods with kernel-based algorithms, although the effectiveness of sparse regularization in neural decoding has been demonstrated both empirically[14–16] and theoretically[17]. Finally, the kernel-based algorithms can evaluate only the distances among the matrices of DMs for each trial, preventing the evaluation of the characteristics of the signals in each trial.

Here, we solved the above three problems by designing a nonlinear feature mapping corresponding to the projection kernel. Known as the kernel trick, kernel-based machine learning algorithms with a kernel function $k(\bullet, \bullet)$ are equivalent to linear machine learning algorithms with a nonlinear feature mapping function $\psi(\bullet)$ satisfying:

$$k(\mathbf{x}_1, \mathbf{x}_2) = \psi(\mathbf{x}_1)^{\mathrm{T}} \psi(\mathbf{x}_2) \ (\forall \mathbf{x}_1, \mathbf{x}_2 \in V)$$

where $V$ is the input space of the kernel function and $\mathbf{A}^{\mathrm{T}}$ denotes transpose of $\mathbf{A}$. In this study, we obtained such a nonlinear feature mapping function $\psi(\bullet)$ for the projection kernel to acquire vectorized features, which can be used in linear machine learning algorithms. In this work, the effectiveness of our proposed features was empirically evaluated using two types of ECoG signals: during hand movements and during visual stimuli. Furthermore, the properties of the sDM features were compared with those of the power features to characterize the properties of the features.

## Results
### Nonlinear feature mapping equivalent to the projection kernel for DMD

The nonlinear feature mapping $\psi(\bullet)$ corresponding to the projection kernel and L2-regularized SVM was obtained as follows. First, the prediction $y$ of the kernel-based classifier for test sample $\mathbf{x}$ is defined as shown in Eq. (3), where $k$, $\mathbf{x}_n$, $a_n$, and $b$ denote the kernel function, $n$th training sample, weight, and bias, respectively:

$$y(\mathbf{x}) = \mathrm{sign}\left(\sum_{n=1}^{N} a_n k(\mathbf{x}_n, \mathbf{x}) + b\right) \quad (3)$$

The projection kernel $k_p$ between the two matrices of DMs for the $i$th and $j$th trials ($\boldsymbol{\Phi}^i$ and $\boldsymbol{\Phi}^j$, respectively; $\boldsymbol{\Phi}^i, \boldsymbol{\Phi}^j \in \mathbb{C}^{P \times K}$) is written as follows

A support vector machine (SVM) was then applied to the acquired Gram matrix of $k_p(\boldsymbol{\Phi}^i, \boldsymbol{\Phi}^j)$ to infer one of three types of movements (cf. Fig. 2a). The SVM model successfully classified the ECoG signals with an accuracy superior to that of a model using power features estimated based on the same ECoG signals. Interestingly, the accuracy of this DM-based decoding approach significantly decreased when the method was applied to ECoG signals with shuffled phases. The higher accuracy than the power features and phase-shuffled ECoG signals indicate that the DM-based decoding method efficiently utilizes the information encoded in the spatiotemporal patterns of the ECoG signals.

Although the DM-based decoding method improved the accuracy of the neural decoding results, three remaining problems must be addressed: (1) the large computational time prevents real-time application of the decoding method in BCIs; (2) the method is incompatible with non-kernel-based machine learning algorithms; and (3) the characteristics of the signals contributing to decoding cannot be easily interpreted. The computational

with $\mathbf{A}^{\dagger}$ denoting conjugate transpose of $\mathbf{A}$:

$$k_p(\boldsymbol{\Phi}^i, \boldsymbol{\Phi}^j) = \left\| \boldsymbol{\Phi}^{i\dagger} \boldsymbol{\Phi}^j \right\|_F^2 = \mathrm{tr}\left[\left(\boldsymbol{\Phi}^i \boldsymbol{\Phi}^{i\dagger}\right)\left(\boldsymbol{\Phi}^j \boldsymbol{\Phi}^{j\dagger}\right)\right]$$
$$= \mathrm{vec}\left(\boldsymbol{\Phi}^i \boldsymbol{\Phi}^{i\dagger}\right)^{\dagger} \mathrm{vec}\left(\boldsymbol{\Phi}^j \boldsymbol{\Phi}^{j\dagger}\right) \qquad (4)$$
$$= \psi\left(\boldsymbol{\Phi}^i\right)^{\dagger} \psi\left(\boldsymbol{\Phi}^j\right)$$

where

$$\psi(\boldsymbol{\Phi}) = \mathrm{vec}(\boldsymbol{\Phi}\boldsymbol{\Phi}^{\dagger}) \qquad (5)$$

Here, we call $\boldsymbol{\Phi}\boldsymbol{\Phi}^{\dagger}$ in Eq. (5) as the spatial DM (sDM) features. Using the vectorized $\boldsymbol{\Phi}\boldsymbol{\Phi}^{\dagger}$ ($\psi(\boldsymbol{\Phi})$), Eq. (3) can be rewritten as follows:

$$y(\mathbf{x}) = \mathrm{sign}\left(\sum_{n=1}^{N} a_n \psi(\boldsymbol{\Phi}^n)^{\dagger} \psi(\mathbf{x}) + b\right) \qquad (6)$$

By setting $\mathbf{w}^{\mathrm{T}} = \sum_{n=1}^{N} a_n \psi(\boldsymbol{\Phi}^n)^{\dagger} = \sum_{n=1}^{N} a_n \psi(\boldsymbol{\Phi}^n)^{\mathrm{T}}$, Eq. (6) can be written as follows, which is the formula for predicting $y$ with a non-kernel-based (linear) classifier based on test sample $\mathbf{x}$:

$$y(\mathbf{x}) = \mathrm{sign}\left(\mathbf{w}^{\mathrm{T}} \psi(\mathbf{x}) + b\right) \qquad (7)$$

Therefore, by using vectorized sDM features obtained by the feature mapping $\psi(\bullet)$ from the DMs, a prediction equivalent to Eq. (3) can be performed with a linear classifier. The sDM features have three interesting characteristics: (1) Although the matrix of DMs $\boldsymbol{\Phi}$ includes complex numbers, the sDM features are always real values because pairs of complex conjugate modes are always obtained by DMD (e.g., Fig. 1d). (2) The sDM features are always symmetrical due to the definition ($\boldsymbol{\Phi}\boldsymbol{\Phi}^{\dagger}$). (3) The sDM features are formulated as the matrix $P \times P$, which corresponds to all combinations of two channels in $P$-channel spatiotemporal signals, such as ECoG signals.

Figure 1e demonstrates $\boldsymbol{\Phi}\boldsymbol{\Phi}^{\dagger}$ for DMs in $[\boldsymbol{\varphi}_1 \ldots \boldsymbol{\varphi}_4]$ ($\boldsymbol{\Phi}\boldsymbol{\Phi}^{\dagger} \in \mathbb{R}^{P \times P}$; $P$ is the number of observation points (e.g., channels of ECoG signals) in the original spatiotemporal signals $\mathbf{x}(t)$. It is worth mentioning that these sDM features are composed of all DMs; thus, the resultant sDM features were determined by modes with frequencies distributed widely in the range of $[0, (1/2\Delta t)]$ Hz. The sDM features for certain frequency bands can also be obtained by selecting modes based on the frequency band (frequency-filtered sDM features; $\boldsymbol{\Phi}\boldsymbol{\Phi}^{\dagger}_{f_{low} < f < f_{high}}$). For example, if the DMs are divided by their frequencies with a threshold of 10 Hz, the frequency-filtered sDM features $\boldsymbol{\Phi}\boldsymbol{\Phi}^{\dagger}_{f < 10Hz}$ and $\boldsymbol{\Phi}\boldsymbol{\Phi}^{\dagger}_{f \geq 10Hz}$ represent the sDM features corresponding to oscillations at two different frequencies (8 and 13 Hz, as shown in Fig. 1f).

## sDM features of ECoG signals during hand movements

The characteristics of the sDM features were evaluated by using the same dataset of ECoG signals from our previous study[10]. This dataset is composed of ECoG signals that were recorded at 1 kHz while 11 patients performed three types of movements with their hand contralateral to the implanted electrodes. Due to clinical requirements, all these patients had subdural electrodes implanted in cortical areas, including the sensorimotor cortex. The dataset consists of ECoG signals from the frontal and parietal cortices (ECoG dataset of arm motor task; Supplementary Table 1). Following the method used in our previous study, the DMs were calculated based on the 500-ms ECoG signals after the cue to start the movements, with truncation of the SVD components; the truncation was performed based on the singular values so that the SVD components with the largest singular values were included in the DM calculation. Because the number of included SVD components, referred to as the rank in this study, largely affects the decoding accuracy, it is important to optimize the rank parameter. However, here, the rank was fixed to 300, which was the same value used in our previous study, to show the representative sDM features and to compare the classification

accuracy and computational time. With the exception of this analysis, the rank parameter was optimized simultaneously with the parameter to train the decoding model via nested cross-validation for all other analyses in this paper (see "Methods").

Figure 2a shows a representative example of the Gram matrix acquired by the projection kernel applied among the DMs from 120 trials, in which patient 1 performed grasping, opening, or pinching movements with his left hand. The Gram matrix showed that the kernel values became similar among trials for the same movement type. For the same DMs from 120 trials, the sDM features ($\boldsymbol{\Phi}\boldsymbol{\Phi}^{\dagger}$: $(\boldsymbol{\Phi}\boldsymbol{\Phi}^{\dagger})_{i,j} \in \mathbb{R}, i,j = 1, \ldots, P$) were acquired for each trial, where $P$ is the number of analyzed channels for the patient. Figure 2b shows an example of the sDM features averaged for each movement type. Because the sDM features are always symmetric, the independent components are on the diagonal and one side of the off-diagonal. We refer to these independent diagonal and off-diagonal components of the sDM features as spatial node DM (snDM) features ($(\boldsymbol{\Phi}\boldsymbol{\Phi}^{\dagger})_{i,i}, i = 1, \ldots, P$) and spatial edge DM (seDM) features ($(\boldsymbol{\Phi}\boldsymbol{\Phi}^{\dagger})_{i,j}, i, j = 1, \ldots, P, i<j$), respectively. When one-way analysis of variance (ANOVA) was applied for each component in the matrix for the three types of movements, the $F$ values of the ANOVA were higher for the diagonal components of the sDM features (snDM features) than for the off-diagonal components of the sDM features (seDM features; Fig. 2c). It was demonstrated that the snDM features had higher selectivity than the seDM features. Moreover, these snDM features and the corresponding $F$ values exhibited high values around the sensorimotor cortex (Fig. 2d, e).

To characterize the neurophysiological properties of the snDM features, the snDM features were compared with the power spectrum density (PSD) features of the same ECoG signals. Because each component of the snDM features corresponds to a channel, the snDM features of all trials were concatenated for each patient to calculate Pearson's correlation coefficient, with the PSD of each frequency concatenated for all channels and trials. The correlation coefficients were Fisher $z$-transformed and averaged among all patients; the resulting coefficients became high from ~80 Hz to ~200 Hz (Fig. 2f), a range that interestingly includes the high-$\gamma$ band (80–150 Hz), which is known to be the most informative frequency band for movement classification[18]. Hence, the results showed that the snDM features were most similar to the high-$\gamma$ power features among the frequency band powers in the ECoG signals. On the other hand, when the snDM features were compared against different trials of the same movement, considering all possible pairs for each patient, the reproducibility of the snDM features was significantly higher than that of the power features, including the high-$\gamma$ power features (Fig. 2g; $p < 0.001$, $F(4,50) = 94.95$, one-way ANOVA; snDM features vs. other features, $p < 0.001$, post hoc Tukey–Kramer test; for reproducibility during visual perception, see Supplementary Fig. 1). These results suggested that snDM features capture similar cortical activities represented by high-$\gamma$ power features with higher reproducibility.

## Neural decoding using sDM features of ECoG signals during hand movements

To assess the feasibility of using sDM features for neural decoding, we compared the computational times and accuracies in classifying the movement types using the Gram matrix of the DMs and the sDM features by SVM. First, the computational times were compared between the kernel-based L2-regularized SVM, with the Gram matrix acquired based on the DMs by the projection kernel, and the non-kernel-based (linear) L2-regularized SVM with the corresponding sDM features. We assessed the decoder training time and the time for the decoder to predict a new sample by changing the number of training samples per class (movement type). Because SVD was a common process among the different decoding methods, the measurement was performed based on the precomputed SVD components. Moreover, it should be noted that the rank parameter to calculate the sDM features was fixed at 300.

The training time of the decoder using the Gram matrix increased exponentially with the number of training samples ($\sim O(n^{1.99})$, where n is number of samples per class; shown as a green line in Fig. 3a). In contrast,

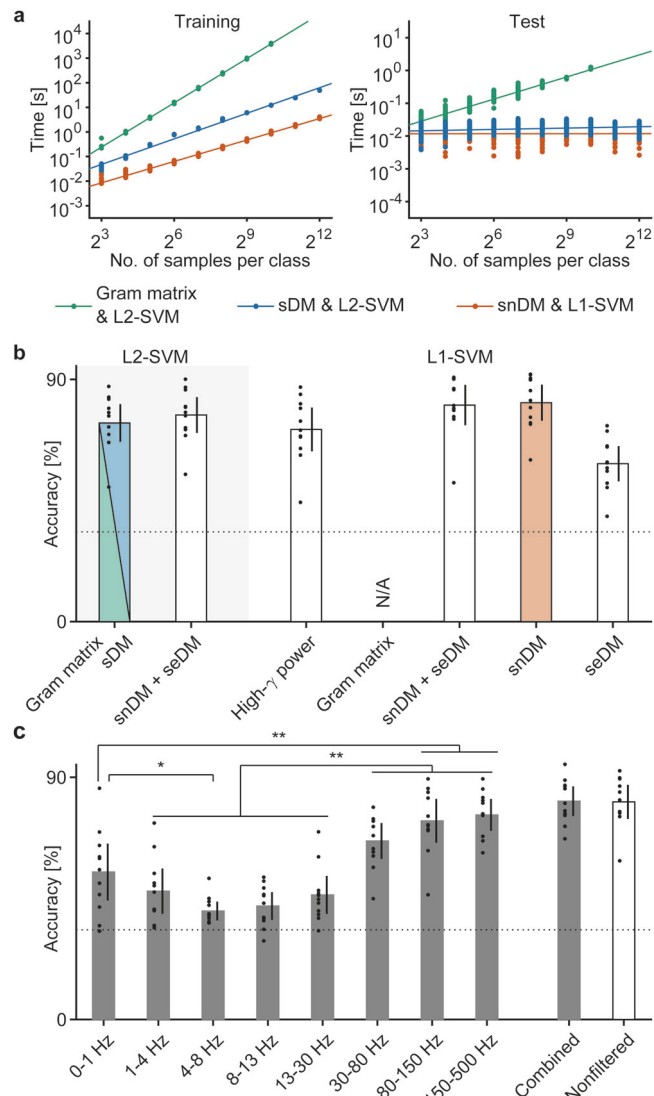

**Fig. 3 | Decoding accuracy and computational time using sDM features.**
**a** Training and testing times of SVM models plotted against different numbers of training samples per class. The training time versus the number of samples per class was fitted by a linear model in log space for both time and number of samples per class to estimate the computational complexity. The computational time of each measurement is shown as dots. **b** Accuracies to classify three types of movements using snDM features, seDM features, and the combination of both features are shown as bars with the 95% CI among the subjects, with individual accuracy represented by dots. For the combination of the SVM model and features shown in (**a**), same color to (**a**) was used to plot the bar; for other combinations, white bars were used. Since the sDM features were constructed such that classification with a linear L2-SVM model based on the sDM features was mathematically equivalent to classification with kernel-based L2-SVM based on the Gram matrix of DMs, the classification accuracies were exactly the same. **c** Frequency-filtered snDM features were calculated for each frequency band of the ECoG signals to perform classification with the L1-SVM model with an optimized rank. Classification was also performed based on the features created by concatenating all of the frequency-filtered snDM features (combined) and snDM features without frequency filtering (nonfiltered). The average classification accuracies are shown by bars with 95% CIs among the patients, with individual accuracies shown as dots. The differences in the classification accuracies among the frequency bands were evaluated by one-way ANOVA with a post hoc Tukey–Kramer test. *$p < 0.05$, **$p < 0.01$.

when the sDM features were used with the linear L2-SVM, the computational complexity was reduced to $\sim O(n^{1.15})$ (shown as a blue line in Fig. 3a). Similarly, the prediction time for a new sample increased with the number of training samples ($\sim O(n^{0.75})$) for the Gram matrix, while the prediction time

using the sDM features and linear L2-SVM was much shorter and increased much more slowly with the number of training samples ($\sim O(n^{0.05})$) (for a comparison of the computational times between high-γ power features and sDM features, see Supplementary Fig. 2). In addition, the decoding accuracies were exactly the same between the method using the Gram matrix and the method using the sDM features (73.80 ± 7.04% (mean ± 95% CI) for both, Fig. 3b). Because the matrix of the sDM features is symmetric, the lower triangular part of the sDM features was redundant for the classification analysis. By performing classification using only the diagonal and upper triangular part of the sDM features (snDM features and seDM features, respectively), the classification accuracy was slightly improved to 76.75 ± 6.67% (labeled as snDM + seDM in Fig. 3a). Use of the sDM features increased the training and testing speeds for the neural decoding process without decreasing the decoding accuracy.

An SVM with L1 regularization (L1-SVM) was then applied to the combined snDM and seDM features. For comparison, we also applied the L1-SVM to the high-γ power features. It is worth mentioning that the L1-SVM cannot be applied to the Gram matrix. The classification accuracy of the combined snDM and seDM features with the L1-SVM (80.45 ± 7.52%) was significantly higher than that of the high-γ power features with the L1-SVM (71.40 ± 8.14%; $p < 0.01$, two-tailed paired $t$ test, $t(10) = 4.61$; Fig. 3b) and that of the Gram matrix of the DMs with the L2-SVM ($p < 0.01$, $t(10) = 4.66$). In addition, the classification accuracy using the snDM features (81.33 ± 6.71%) was similar to the accuracy using the combined snDM and seDM features (Fig. 3b). In contrast, the accuracy using the seDM features was lower (58.67 ± 6.55%). Notably, the use of the snDM features with the linear L1-SVM reduced the training time to $\sim O(n^{0.97})$ and the prediction time to $\sim O(n^{0.001})$ compared to the use of the sDM features with the L2-SVM (Fig. 3a). Therefore, these results demonstrated that the sDM features, especially the snDM features, with the L1-SVM model improved the classification accuracy and computational time for the neural decoding of ECoG signals recorded during hand movements (for the specific effects of the available electrodes on the classification accuracy and the importance of each electrode, see Supplementary Figs. 3 and 4).

As previously mentioned, the sDM features were composed of DMs from the full frequency range; hence, the sDM features were unlikely to capture differences in the frequencies of the DMs. To assess the differences in the frequencies of the modes for different movement types, the classification accuracy was evaluated using snDM features calculated from DMs whose frequencies fell within a given frequency range (Fig. 1f; for visualization of the frequency-filtered snDM features, see Supplementary Fig. 5). Here, the evaluation was performed with the L1-SVM model for conventional frequency bands (0–1, 1–4, 4–8, 8–13, 13–30, 30–80, 80–150, and 150–500 Hz). The classification accuracies significantly differed among the frequency bands ($p < 0.001$, $F(7,80) = 19.34$, one-way ANOVA); interestingly, the classification accuracies for the frequency bands of 80–150 Hz and 150–500 Hz, which are known to be informative for movement decoding using power features[18], were significantly higher than those of the other frequency bands except 30–80 Hz ($p < 0.01$, post hoc Tukey–Kramer test of one-way ANOVA). However, the classification accuracy using the combined frequency-filtered snDM features from all bands was similar to that using the (nonfiltered) snDM feature (combined frequency-filtered snDM features, 81.12 ± 5.54%; snDM features, 80.88 ± 6.34%). Notably, the classification accuracy did not improve when frequency-filtered seDM features were included (combined frequency-filtered seDM features, 65.17 ± 8.87%; combined frequency-filtered snDM and seDM features, 79.39 ± 6.09%). These results suggested that (1) the information about the movement type included in each frequency band was not complementary, and (2) the differences in the movement types did not largely affect the frequencies of the DMs.

## Comparison of neural decoding accuracy for different types of tasks and decoding using ECoG signals
The accuracies of neural decoding using snDM and seDM features based on ECoG signals were compared among different types of tasks and with the

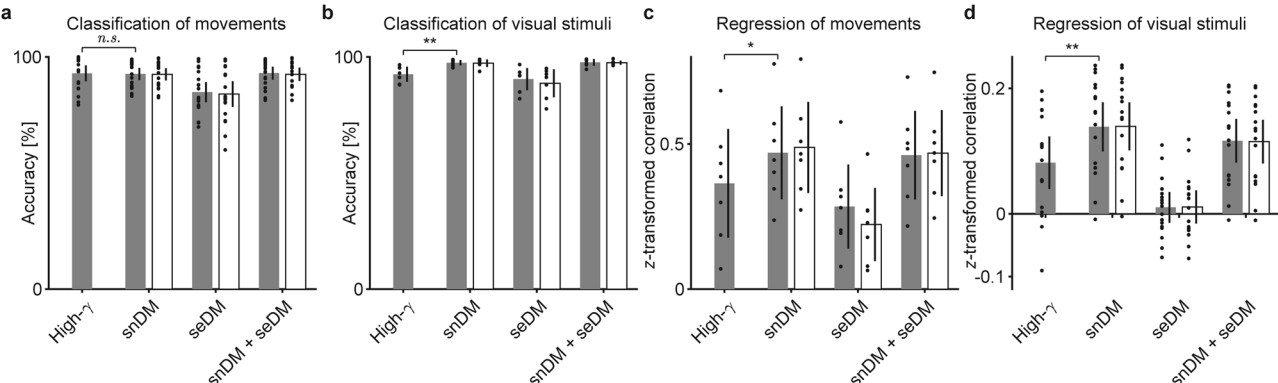

**Fig. 4 | Neural decoding accuracies for various ECoG datasets.** Bars show decoding accuracies using high-γ power features, snDM features, seDM features, and the combination of both features (black bars) and those using the corresponding part of the frequency-filtered sDM features (white bars) for the ECoG datasets: **a** hand versus tongue movement task ($n = 19$), **b** image perception task ($n = 7$), **c** finger flexion task ($n = 7$; the average of the Fisher $z$-transformed correlation coefficients for the thumb, index finger, and little finger is shown, following Miller et al.[32]; for the correlation coefficients for each patient and each finger, see Supplementary Table 4), and **d** video perception task ($n = 17$; the average of the Fisher $z$-transformed correlation coefficients among 1000 dimensions is shown). Error bars represent 95% CIs among patients, with dots representing decoding accuracy for each patient. For classification in (**a**) and (**b**), L1-SVM was applied. The differences in the decoding accuracies using the high-γ power features and snDM features were evaluated by means of two-tailed paired $t$ tests. *$p < 0.05$, **$p < 0.01$.

accuracies using high-γ power features. Here, we used our own ECoG dataset of video perception task (Fukuma et al.[19]) and an open dataset of ECoG signals, including hand versus tongue movement task, finger flexion task, and image perception task (Miller[20]). Neural decoding by either classification or regression was performed for each of these datasets. As summarized in Fig. 4, the proposed method successfully decoded ECoG signals with an accuracy higher than or comparable to that using high-γ power features regardless of task or decoding type (due to the computational time, L2 regularization was used for regression throughout this study; Supplementary Fig. 6 and Supplementary Tables 3, 4). Consistent with the results based on the ECoG dataset of the arm motor task, the snDM features enabled neural decoding with higher accuracy than the seDM features for all datasets (Fig. 4, black bars), and use of the frequency-filtered snDM and seDM features did not show considerable improvement in the decoding accuracy (Fig. 4, white bars).

## Discussion

We proposed sDM features to characterize spatiotemporal signals. The sDM features were DM representations transformed by a mathematical conversion equivalent to the projection kernel and kernel-based L2-SVM. Throughout this study, the following advantages of the sDM features were shown. (1) The sDM features enable neural decoding with DMs for real-time BCIs that require small delays to control external devices based on ECoG signal changes. In fact, the use of the sDM features drastically reduced the prediction time from $\sim O(n^{0.75})$ to $\sim O(n^{0.001})$ ($n$ is number of training samples), which is the time achieved to classify the Gram matrix acquired by the projection kernel based on the DMs. Notably, the training time of the decoder was also reduced from $\sim O(n^{1.99})$ to $\sim O(n^{0.97})$, enabling the use of more training samples. (2) The sDM features allow neural decoding to be performed with L1 regularization, thereby improving the classification accuracy using ECoG signals. Moreover, even with regression with L2 regularization, the regression accuracy for the finger flexion task increased. These results strongly suggest that the sDM features are promising for fast and accurate decoding. (3) The characteristics of the signals can be inferred based on the different behaviors of the snDM, seDM, and the frequency-filtered sDM features. Based on the results, the ECoG signals were characterized by snDM features as opposed to seDM features. Furthermore, because the use of frequency-filtered sDM features did not show considerable improvement in the decoding accuracy compared to use of the nonfiltered sDM features, the frequency information appears to be less effective in the neural decoding of ECoG signals. Therefore, the proposed sDM features have several key factors that demonstrate their effectiveness as

BCI decoders for ECoG signals: (1) high accuracy, (2) fast computational speed, (3) good scalability, and (4) good interpretability.

The high-γ activities in ECoG signals have been determined to be the most informative frequency band for neural decoding[18,19,21–24], reflecting the spiking activities of neurons[25]; hence, the use of high-γ powers has been a standard method to extract useful information from cortical activity. On the other hand, the (nonfiltered) sDM features proposed in this study are constructed based on DMs regardless of their widely distributed frequencies. Interestingly, although the high-γ powers and the sDM features are produced in different ways, the correlation coefficient between the high-γ powers and the snDM features suggests that they extract similar information. Moreover, the snDM features have smaller variance within the same task than the high-γ powers, which likely contributes to the improvement in the decoding accuracy. It is worth noting that the use of the frequency-filtered snDM features did not lead to a considerable improvement in the decoding accuracy of the ECoG signals. Taken together, the results suggest that the motor and visual information in the ECoG signals is not encoded in the frequencies of the DMs, although the high-γ power features are highly informative among the various frequency bands. The sDM features are novel electrophysiological features that stably extract neural information without explicitly selecting the frequency band for neural decoding.

The proposed method utilizes information about the nonlinear dynamics in ECoG signals while avoiding the computational costs incurred for decoding via kernel methods. In our previous study, decoding was performed based on a symmetric positive definite matrix (Gram matrix) calculated by means of a projection kernel on a Riemannian manifold; in contrast, the proposed method directly converts the DMs into sDM features, enabling the use of non-kernel-based decoders (e.g., L1-regularized SVM). It should also be mentioned that both DMD and traditional empirical mode decomposition (EMD) offer effective ways to derive information about the nonlinear dynamics in time-series data. However, DMD has the following main advantages over EMD for high-dimensional data with nonlinearities, such as in the present case. (1) The decompositions obtained by DMD are known to be closely related to several physical concepts, such as phase reduction as discussed in the field of nonlinear science, allowing the extracted information to be interpreted both physically and mathematically. (2) EMD is a method that captures nonlinearity and nonstationarity in data based on the extraction and smoothing of local extreme values in the data, and in principle, it is known to be numerically unstable and generally difficult to apply to high-dimensional data due to its large computational cost. On the other hand, DMD is based on relatively stable numerical computations (SVD and eigenvalue decomposition), is known to be robust

for high-dimensional data, and is capable of extracting information that captures global dynamic features in data[6]. In fact, the proposed sDM features are informative for various types of tasks and decoding (Fig. 4). Considering the improved computational cost in this study, the proposed method may also be effective even for different modalities of spatiotemporal signals with high spatial dimensions.

Finally, the proposed sDM features can be further combined with different machine learning algorithms other than simple SVM and regression. Many new algorithms have been proposed for decoding ECoG signals, including methods based on deep neural networks (DNNs) with long short-term memory[26], recurrent neural networks[21], and gradient boosting trees[27]. In fact, it has been reported that the decoding accuracy (correlation coefficient) for the finger flexion task can reach 0.50 for all fingers and for all patients in a dataset with gradient boosting trees[27], whereas the accuracy was 0.37 for snDM features with L2-regularized regression (see Supplementary Table 4). Thus, the proposed sDM features may improve the decoding accuracy when combined with these algorithms. Moreover, recent studies using DNNs have shown the viability of across-patient decoding by means of weights for individual patients (subject blocks)[28] or electrode-level data projections onto predefined brain regions[29]. With these techniques, sDM features, which capture the spatiotemporal patterns of multiple signals, could similarly be used for decoding across patients.

In conclusion, our proposed sDM features were demonstrated to be effective for the fast and accurate decoding of ECoG signals in various tasks. Furthermore, snDM features without frequency band selection appear to be the most effective features for decoding ECoG signals.

## Methods
### Analyzed datasets
This study employed two in-house datasets that were used in our previous reports (ECoG dataset of arm motor task (Shiraishi et al.[10]) and ECoG dataset of video perception task (Fukuma et al.[19])), and publicly available datasets (ECoG signals acquired during hand versus tongue movement, flexion of fingers, and visual perception of face and house images (Miller[20])). To record the in-house datasets, experiments were performed in accordance with the experimental protocol approved by the ethics committee of each hospital (Osaka University Medical Hospital: Approval No. 08061, No. 14353, No. 19257, UMIN000017900; Juntendo University Hospital: Approval No. 18-164; Nara Medical University Hospital: Approval No. 2098). Prior to the experiments, all subjects or their guardians provided written informed consent to participate in the study. All ethical regulations relevant to human research participants were followed.

### ECoG dataset of the arm motor task (in-house dataset from Shiraishi et al.[10])
**Subjects.** The ECoG dataset of the arm motor task consisted of eleven subjects (7 males; age range, 13–66 years), with subdural electrodes placed on their front-parietal area, including the sensorimotor cortex. All subjects were implanted with intracranial electrodes prior to the study for the purpose of treating their drug-resistant epilepsy.

**Task procedure.** The subjects were instructed to perform three types of movement with their upper limb contralateral to the sensorimotor cortex where the subdural electrodes were implanted. Three types of movement were selected among grasping, pinching, hand opening, thumb flexion, and elbow flexion[18] according to their performance ability and comfort. For each trial, three visual and auditory cues were provided at intervals of 1 s; at the timing of the last cue, the subjects performed one of the three types of movement once and returned to the resting position, relaxing their hands or elbows with slightly flexed joints. For the types of performed movements and number of trials for each movement type, see Supplementary Table 1.

**Experimental settings and ECoG recordings.** The subjects were seated on chairs to perform the movement tasks. A computer screen was placed

in front of the subjects to show the movement cue, which was also delivered auditorily. The presentation of the cues was controlled using ViSaGe (Cambridge Research System, Rochester, UK). During the experiment, ECoG signals were recorded at 1 kHz by EEG-1200 (Nihon Koden, Tokyo, Japan) by referencing the average of two intracranial electrodes. Digital pulses denoting the timing of the cue were recorded synchronously with the ECoG signals.

**Signal preprocessing.** Signal preprocessing of this dataset was performed as described in our previous study[10] by rejecting noisy channels and channels located outside of the front-parietal area via visual inspection (for the number of channels in the dataset, see Supplementary Table 1). For the analyses in this study, the ECoG signals were common-average referenced and cropped from 0 to 500 ms with respect to the movement cue.

**Division of the dataset for classification.** This dataset was evaluated by classification analysis with nested cross-validation for each patient. To accurately estimate the classification accuracy, 10-fold outer cross-validation was repeated 10 times by changing the division of the dataset to calculate the average of the 10 classification accuracies. For each outer fold, inner cross-validation was also repeated 10 times by changing the division of the samples to accurately estimate the best decoder parameters. The imbalance between the number of samples for each label (movement type) was minimized for each division.

### ECoG dataset of the video perception task (in-house dataset from Fukuma et al.[19])
**Subjects.** The ECoG dataset for the video perception task consisted of 17 subjects (12 males; age range 11–51 years), with subdural electrodes placed around their visual and temporal cortices for the treatment of epilepsy. One subject participated twice within 2 years due to a second surgery (E07 and E11).

**Task procedure.** All seventeen subjects (E01–E17) were shown the same six 10-min videos (training videos), and 12 subjects (E01, E03, E06, E07, and E09–E16) were also shown another 10-min video (validation video). No fixation point was presented in the video stimuli; the subjects were instructed to freely watch the videos. The presentation of the training videos took 1–3 days to complete. The validation video was presented after the presentation of all training videos.

**Visual stimuli.** The six training videos and the validation video were created by sequentially concatenating short film or animation clips. The clips were cutouts from one of 75 trailers or behind-the-scene features downloaded from Vimeo and had a median duration of 16 s (interquartile range, 14 to 18 s). The six 10-min training videos were created by concatenating 224 clips, and the 10-min validation video was created with four repetitions of a 2.5-min video composed of 11 clips. The short video clips were cut so that they did not overlap; hence, there were no overlapping scenes not only between the training videos and the validation video but also among the training videos. The resulting videos contained scenes that widely varied in semantic meaning, such as animals, foods, landscapes, and text.

**Construction of the semantic vectors: training the skip-gram model.** A skip-gram model was trained using Japanese Wikipedia dump data with the following steps based on the procedure described in a study by Nishida and Nishimoto[30]. (1) Words were segmented and lemmatized from Japanese text in the articles in the Wikipedia dump to create a text corpus using MeCab[31], an open-source text segmentation software, along with the Nara Institute of Science and Technology (NAIST) Japanese dictionary, a vocabulary database for MeCab. (2) In the text corpus, words other than nouns, verbs, and adjectives and words that appeared less than 120 times were discarded, resulting in a text corpus of

365,312,470 words, consisting of 94,337 nouns, 4922 verbs, and 631 adjectives. (3) By using the Gensim Python library, a skip-gram model was trained with the text corpus. The training parameters were set as follows: dimension of word vector representation, 1000; window size, 5; number of negative samples, 5; use of hierarchical softmax function, no.

**Construction of the semantic vectors: conversion to the semantic vectors.** For each 1-s scene in the training videos and validation video, the semantic meaning of the scene was represented as a semantic vector based on the scene annotations and the trained skip-gram model. A still image was extracted from each 1-s scene, resulting in 3600 images for the six 10-min training videos and 150 images for the first 2.5 min of the validation video. Each extracted image was manually annotated by five annotators with descriptive sentences containing 50 or more Japanese characters. Using the same preprocessing method performed with the Japanese Wikipedia dump data, lemmatized words were extracted from the annotations and filtered by discarding words that did not exist in the text corpus of the trained skip-gram model. The remaining words were then converted to 1000-dimensional vectors using the trained skip-gram model, which were first averaged within each annotation and then averaged among the five annotators to create a 1000-dimensional semantic vector for each scene.

**Experimental settings and ECoG recordings.** The subjects either sat on beds in their hospital rooms or were seated on chairs to perform the experimental tasks. A computer screen was placed in front of the subjects to show the video stimuli. A pair of speakers was also placed near the subjects to play sounds during the presentation of the video stimuli. During the experiment, ECoG signals were recorded at 10 kHz by EEG-1200 (Nihon Koden, Tokyo, Japan) by referencing the average of two intracranial electrodes. The presentation timing of the video stimuli was monitored by DATAPixx3 (VPixx Technologies, Quebec, Canada) and recorded as digital signals synchronized to the ECoG signals.

**Signal preprocessing.** Signal preprocessing of this dataset was performed as described in our previous study[19] by rejecting noisy channels via visual inspection (for the number of channels in the dataset, see Supplementary Table 2). The ECoG signals in the dataset were filtered with a lowpass filter (8th-order Chebyshev Type I infinite impulse response filter) and downsampled to 1 kHz. The downsampled ECoG signals were then rereferenced by common averaging. For the regression analysis, the ECoG signals corresponding to the 1-s scenes were used.

**Division of the dataset for regression.** To enable direct comparison with our previous study, regression was performed with nested cross-validation using the same division of the dataset as in our previous study[19], in which the samples were divided into 10 groups so that the scenes from the same video source were kept in the same group, and the imbalance in the number of trials among the groups was minimized. Hence, nonrepeated 10-fold outer cross-validation with nonrepeated 9-fold inner cross-validation was performed for the regression.

**ECoG dataset of the hand versus tongue movement task ("motor_basic" experiment in the open dataset from Miller[20])**
**Dataset overview.** To acquire the dataset, the patients were implanted with intracranial electrodes around the front-parietal area. The patients performed repetitive movements with their hand (synchronous flexion and extension of all fingers) or tongue (sticking the tongue in and out from their mouth) at their own pace (~1–2 Hz) while movement cues were provided for 2 or 3 s. Each movement type was repeated 15–45 times. ECoG signals were recorded at 1 kHz. Nineteen patients included in the dataset were used for the analysis in this study.

**Signal preprocessing.** For each patient, ECoG signals were rereferenced by common averaging among all channels. For the classification analysis,

ECoG signals from 0 to 2 s with respect to the start of the moment cues were obtained.

**Division of the dataset for classification.** For this dataset, the classification analysis was performed for each subject by nested cross-validation. To accurately estimate the classification accuracy, 10-fold outer cross-validation was repeated 10 times by changing the division of the dataset, and the average of the 10 classification accuracies was calculated. In addition, for each outer fold, 10-fold inner cross-validation was also repeated 10 times by changing the division of the samples to better estimate the decoding parameter. The division was performed so that the imbalance between the numbers of samples for each label (hand or tongue movement) was minimal.

**ECoG dataset of the image perception task ("faces_basic" experiment in the open dataset from Miller[20])**
**Dataset overview.** Patients implanted with intracranial electrodes in the inferotemporal subdural space participated in a visual perception task in which face or house images were presented. During the recording of the ECoG signals at 1 kHz, the patients were presented with luminance- and contrast-matched grayscale face and house images for 400 ms in random order, with an interstimulus interval of 400 ms. In each of the three repeated runs, 50 different face or house images were presented. All fourteen patients in the dataset were included in the analysis for this study.

**Signal preprocessing.** For each patient, rereferencing of ECoG signals was performed by common averaging among all channels. For the classification analysis, ECoG signals from 0 to 400 ms with respect to the image presentation were used.

**Division of the dataset for classification.** Classification analyses with this dataset were performed with a within-patient approach by nested cross-validation. To accurately estimate the classification accuracy, 10-fold outer cross-validation was repeated 10 times by changing the division of the dataset to average the classification accuracies among the repetitions. The decoding parameters of each outer fold were estimated by 10-fold inner cross-validation, which was also repeated 10 times by changing the division of the samples. During the division of the dataset, the number of samples for each label (face or house images) was blanched in each group.

**ECoG dataset of the finger flexion task ("fingerflex" experiment in the open dataset from Miller[20])**
**Dataset overview.** To acquire the dataset, patients were implanted with intracranial electrodes around the front-parietal area. The patients performed repeated movements (flexion and extension) of individual fingers; the movement of each finger was measured at 25 Hz by a 5-DOF data glove with simultaneous recording of 1-kHz ECoG signals. Patients were given a 2-second cue to move individual fingers at their own pace. The movement cue for each finger was presented in a random order, with an intertrial interval of 2 s. There were 30 movement cues for each finger. All nine patients in the dataset were included in the analysis.

**Signal preprocessing.** ECoG signals were first rereferenced by common averaging among all channels for each patient. For each measurement of finger flexion, the corresponding ECoG signals were cropped to form a sample to be regressed with the following procedure: (1) In the dataset, values for finger flexion movements were upsampled from 25 Hz to 1 kHz and saved with the 1-kHz ECoG signals, leading to 40 continuous samples for the same value. Based on these values, the timing of the first sample was identified. The finger flexion values for these samples were selected as the target variables for the later regression analysis. (2) The ECoG signals corresponding to the selected samples were cropped with a 300-ms time window; here, the time window was placed at 84 ± 150 ms

with respect to the selected samples because the original study reported that the best Pearson's correlation coefficient was obtained with an 84 ms offset[32].

**Division of the dataset for regression.** To prevent overestimation of the accuracy, the samples in the dataset for each patient were divided into 10 groups by splitting the time sequence of the samples. Nested cross-validation for the regression analysis was performed with this division for both the inner and outer folds; hence, nonrepeated 10-fold outer cross-validation was performed using nonrepeated 9-fold inner cross-validation.

## DMD

Assuming that the spatiotemporal signals originate from one dynamic system, the system can be described as follows:

$$\frac{d\mathbf{x}}{dt} = f(\mathbf{x}, t; \mu) \quad (10)$$

where $\mathbf{x}(t) \in \mathbb{R}^P$ is a vector representing the state of the dynamic system at time $t$, and $\mu$ and $f(\bullet)$ denote the system parameters and the dynamics, respectively. Considering that the actual signal measurement is performed in discrete time intervals of $\Delta t$, the discrete time representation of the dynamic system corresponding to Eq. (10) can be written as follows:

$$\mathbf{x}_{l+1} = \boldsymbol{F}(\mathbf{x}_l) \quad (11)$$

where $\mathbf{x}_l$ denotes the $l$th measurement of the system ($\mathbf{x}_l = \mathbf{x}(l\Delta t)$; $l = 1, 2, ..., L$). Practically, the dynamics $\boldsymbol{F}$ needs to be estimated from the observed signals; here, the DMD method estimated the dynamics by linear approximation as follows:

$$\mathbf{x}_{l+1} = \mathbf{A}\mathbf{x}_l \quad (12)$$

Then, $\mathbf{A}$ is acquired by minimizing the approximation error $||\mathbf{x}_{l+1} - \mathbf{A}\mathbf{x}_l||_2$ across all measurements of $l = 1, 2, ..., L - 1$.

To minimize the approximation error, two matrices of the measurement, $\mathbf{X}$ and $\mathbf{X}'$, are introduced:

$$\mathbf{X} = [\mathbf{x}_1 \ldots \mathbf{x}_{L-1}],$$

$$\mathbf{X}' = [\mathbf{x}_2 \ldots \mathbf{x}_L].$$

In the original DMD method, the dimension of $\mathbf{X}$ was assumed to be $P \gg L$; for the implementation in this study, see the "Signal stacking" section for a more detailed explanation. The linear approximation in Eq. (12) can be written as $\mathbf{X}' \approx \mathbf{A}\mathbf{X}$, where the optimized $\mathbf{A}$ is given by $\mathbf{A} = \mathbf{X}'\mathbf{X}^+$ and + is the Moore–Penrose pseudoinverse. By applying SVD to $\mathbf{X}$:

$$\mathbf{X} \approx \mathbf{U}\mathbf{S}\mathbf{V}^*$$

where $\mathbf{U} \in \mathbb{C}^{P \times K}$, $\mathbf{S} \in \mathbb{C}^{K \times K}$, $\mathbf{V} \in \mathbb{C}^{L \times K}$, $*$ represents the conjugate transpose, and $K$ denotes the rank used for the SVD approximation. Notably, the left and right singular matrices ($\mathbf{U}$ and $\mathbf{V}$, respectively) satisfy $\mathbf{U}^*\mathbf{U} = \mathbf{I}$ and $\mathbf{V}^*\mathbf{V} = \mathbf{I}$. This process assumes a low-dimensional structure for the dynamics. Here, $\mathbf{A}$ can be obtained by using the pseudoinverse of $\mathbf{X}$ acquired by the SVD:

$$\mathbf{A} = \mathbf{X}'\mathbf{V}\mathbf{S}^{-1}\mathbf{U}^*$$

Because the dimension of the measurement ($P$) is large, eigenvalue decomposition of $\mathbf{A}$ requires considerable computational resources. The DMD method addresses this problem by leveraging the orthogonal matrix

$\mathbf{U}$, yielding:

$$\widetilde{\mathbf{A}} = \mathbf{U}^*\mathbf{A}\mathbf{U} = \mathbf{U}^*\mathbf{X}'\mathbf{V}\mathbf{S}^{-1}.$$

Then, the eigendecomposition of $\widetilde{\mathbf{A}}$ was performed as follows:

$$\widetilde{\mathbf{A}}\mathbf{W} = \mathbf{W}\boldsymbol{\Lambda}$$

where each column in $\mathbf{W}$ is an eigenvector and $\boldsymbol{\Lambda}$ is the diagonal matrix of the corresponding eigenvalues $\lambda_k$. Finally, the approximated eigenvectors of $\mathbf{A}$ (DM) are obtained as the columns in $\boldsymbol{\Phi}$, with the corresponding eigenvalues given by $\boldsymbol{\Lambda}$:

$$\boldsymbol{\Phi} = \mathbf{X}'\mathbf{V}\mathbf{S}^{-1}\mathbf{W}.$$

By introducing the variable $\omega_k = \ln(\lambda_k)/\Delta t$, the original dynamics can be approximated as:

$$\mathbf{x}(t) \approx \sum_{k=1}^K \boldsymbol{\varphi}_k e^{\omega_k t} b_k \quad (13)$$

where $b_k$ is the initial condition of the mode.

Here, $\mathbf{b} = (b_1, , b_K)^T$ can be obtained as $\mathbf{b} = \boldsymbol{\Phi}^+\mathbf{x}(0)$. By rewriting $\omega_k$ in Eq. (13), Eq. (2) can be obtained.

## Signal stacking

The original DMD method was developed for signals with $P \gg L$, where $P$ and $L$ denote the number of recording sites and measurements, respectively. However, for neural signals, $P$ is usually smaller than $L$. In these cases, the signals can be augmented by stacking them $h$ times to create the two measurement matrices $\mathbf{X}$ and $\mathbf{X}'$:

$$\mathbf{X} = \begin{bmatrix} \mathbf{x}_1 & \mathbf{x}_2 & \cdots & \mathbf{x}_{L-h} \\ \mathbf{x}_2 & \mathbf{x}_3 & \cdots & \mathbf{x}_{L-h+1} \\ \vdots & \vdots & \ddots & \vdots \\ \mathbf{x}_h & \mathbf{x}_{h+1} & \cdots & \mathbf{x}_{L-1} \end{bmatrix},$$

$$\mathbf{X}' = \begin{bmatrix} \mathbf{x}_2 & \mathbf{x}_3 & \cdots & \mathbf{x}_{L-h+1} \\ \mathbf{x}_3 & \mathbf{x}_4 & \cdots & \mathbf{x}_{L-h+2} \\ \vdots & \vdots & \ddots & \vdots \\ \mathbf{x}_{h+1} & \mathbf{x}_{h+2} & \cdots & \mathbf{x}_L \end{bmatrix}.$$

Throughout this study, $h$ was the minimum integer that satisfies $h \geq \frac{L+1}{P+1}$. Moreover, out of the $hP$ DMs obtained from these stacked signals, the first $P$ DMs were used for the analysis.

## Acquisition of the Gram matrix and sDM features

DMD was first applied to the preprocessed spatiotemporal signals ($\mathbf{x}(t)$) of each trial in each dataset. Each DM ($\boldsymbol{\varphi}$) in the matrix of DMs ($\boldsymbol{\Phi}$) for the sample was then L2-normalized following the method used in our previous study[10]. A projection kernel was then applied to each pair of the matrix of the L2-normalized DMs to generate the Gram matrix; similarly, according to Eq. (5), the sDM features were calculated based on the matrix of the L2-normalized DMs.

For the frequency-filtered sDM features, the following frequency bands were used to group the DMs: 0–1, 1–4, 4–8, 8–13, 13–30, 30–80, 80–150, and 150–500 Hz. When no DMs were within a band, all components of the frequency-filtered sDM features for the band were set to zero.

## Calculation of the PSD and power features

The PSD and power features were calculated based on the same 500-ms signals ($\mathbf{x}(t)$) that were used to calculate the DMs and sDM features for the ECoG dataset of the arm motor task. For each channel in $\mathbf{x}(t)$, the PSD was

calculated using a Hamming window and fast Fourier transformation of 512 points. To calculate the power features, the PSD was averaged within the given frequency band (e.g., 80–150 Hz for the high-γ band).

## Neural decoding

**Nested cross-validation.** Throughout this study, the training parameters (cost or λ parameter for the decoder, and rank parameter for the sDM features) were always optimized only using the training samples independently from the testing samples to prevent overfitting of the decoder. For all datasets, nested cross-validation was applied; for each outer cross-validation, the testing samples of the outer fold were decoded with a decoder trained based on all training samples (of the outer fold), with the optimized parameters estimated based on the inner cross-validation with the training samples.

**Classification analysis.** In this study, classification analysis was performed with either an L2-regularized SVM or an L1-regularized SVM. For the L2-regularized SVM model decoding based on the Gram matrix, classification was performed by LIBSVM 3.1[33] with the following parameters: svm_type, 0 (C-SVC); kernel_type, 4 (precomputed kernel). For the L2-regularized SVM with the linear kernel, the following parameters were used by LIBSVM: svm_type, 0 (C-SVC); kernel_type, 0 (linear). For the L1-regularized SVM, the classification was performed by LIBLINEAR 1.8[34] with the following parameters: s, 6 (L1-regularized logistic regression). In each case, the other parameters were set to their default values. For all classification analyses, the cost for the SVM was optimized by (nested) cross-validation from candidates of $10^{-1}$, $10^{0}$, $\cdots$, $10^{8}$. When the number of training samples for each class was imbalanced, the samples for the classes with less samples were repeatedly included so that the number of samples was increased to that of the class with the most samples. Moreover, the classification accuracies were evaluated by the balanced accuracy.

**Regression analysis.** Due to the limitation of the computational time, L2-regularized ridge regression was used in this study. Parameter λ was optimized from candidates of $10^{-8}$, $10^{-7}$, $\cdots$, $10^8$ by (nested) cross-validation for each dimension of the dependent variables. The optimization was performed by minimizing the mean square error, and the regression accuracy was evaluated based on the average of the correlation coefficients between the true and predicted values for each dimension.

## Evaluation of computational time

Decoder training time and decoder testing time on a new sample were assessed by varying the number of samples per class ($n$) using the precomputed DMs of patient 1 in the arm motor task. First, trials were randomly resampled so that the number of trials in each class was equal to $n$, and then the DMs of the resampled trials were used to train the decoder model while training time was measured. The cost parameter for training was selected as the most frequent value among the optimized costs in the outer folds of the nested cross-validation to calculate the classification accuracy. To measure the testing time of the decoder, the precomputed DMs of a randomly selected trial were applied to the decoder. The measurement was repeated 100 times by changing the seed value for the random number generator.

## Statistics and reproducibility

The reproducibility of the snDM features and the power features was tested by one-way ANOVA with post hoc Tukey–Kramer tests (Fig. 2g).

The classification accuracy of the L1-regularized SVM with combined snDM and seDM features was compared with that of the L1-regularized SVM with high-γ power features by two-tailed paired *t* tests (Fig. 3b).

The classification accuracies using the frequency-filtered snDM features were tested among the frequency bands by one-way ANOVA with post hoc Tukey–Kramer tests to determine the frequency band that was most informative for classification (Fig. 3c).

The decoding accuracies using the high-γ power features and the snDM features were compared using a two-tailed paired *t* test (Fig. 4).

The reproducibility of the proposed method was verified on two in-house datasets and three open datasets.

### Reporting summary

Further information on research design is available in the Nature Portfolio Reporting Summary linked to this article.

## Data availability

The source data for the graphs in this paper are provided in Supplementary Data 1. Other relevant data are available under a formal data sharing agreement.

## Code availability

The code used in this study are publicly available on github (https://github.com/yanagisawa-lab/fast-accurate-and-interpretable-decoding-of-electrocorticographic-signals-using-DMD).

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

## Acknowledgements

This research was supported by the Japan Science and Technology Agency (JST) Core Research for Evolutional Science and Technology (CREST) (JPMJCR18A5), JST Precursory Research for Embryonic Science and Technology (PRESTO) (JPMJPR2128), JST Exploratory Research for Advanced Technology (ERATO) (JPMJER1801), JST Moonshot R&D (JPMJMS2012), JST MEXT Q-LEAP (JPMXS0120330644), Acquisition, Technology & Logistics Agency (ATLA) Innovative Science and Technology Initiative for Security (JPJ004596), Japan Agency for Medical Research and Development (AMED) (JP19dm0207070, JP19dm0307103, and JP23dm0307009), Japan Society for the Promotion of Science (JSPS) Grant-in-Aid for Early-Career Scientists (22K15623), and JSPS Grants-in-Aid for Scientific Research (KAKENHI) (20K16465).

## Author contributions

Ryohei Fukuma: software, validation, formal analysis, investigation, visualization, writing—original draft, writing—review and editing. Kei Majima: development of formula for sDM features, software, investigation, writing—original draft, writing—review and editing. Yoshinobu Kawahara: conceptualization, methodology, writing—review, and editing. Okito Yamashita: conceptualization, writing—review and editing, funding acquisition. Yoshiyuki Shiraishi: writing—review and editing. Haruhiko Kishima: resources. Takufumi Yanagisawa: conceptualization, investigation, data curation, writing—original draft, writing—review and editing, supervision, project administration, funding acquisition.

## Competing interests

The authors declare no competing interests.
