## [Peer Review File · Communications Biology]

Reviewers' comments:

Reviewer #1 (Remarks to the Author):

Thank you for inviting me to review this interesting manuscript. The present study describes a novel adaptation of dynamical modes for invasive brain signal decoding. The topic is of general relevance and the methods are clearly described. I only have three major points.

1) In addition to the demonstrated comparisons, the authors should compare their results to the best individual channel. This is relevant because for future brain computer interfaces, the number of channels will be very limited and therefore any measure that requires complex and variable spatial locations should clearly outperform a more simple single electrode approach. Additionally, the authors should investigate the performance and stability with relation to number of available channels, e.g. by systematically removing channels. This is relevant to inform the design of brain computer interfaces that could benefit from the approach.

2) The authors should discuss the opportunities for across patient decoding, which is an increasingly recognized problem with some solutions at hand, e.g. see <https://doi.org/10.1088/1741-2552/abda0b> & <https://doi.org/10.21203/rs.3.rs-3212709/v1> .

3) The authors should make an effort to provide interpretability with respect to source location in addition to the frequency band.

Reviewer #2 (Remarks to the Author):

“Fast, accurate, and interpretable decoding of electrocorticographic signals using dynamic mode decomposition” by Fukuma and colleagues seeks to validate a novel method of signal decomposition – dynamic mode decomposition (DM) via spatial DM (sDM) features for neural decoding. They first find that sDMs correlate most highly with higher frequency features (>~50 Hz) of the neural signal. They also find that as compared to features in this frequency range, sDMs show more robust trial-wise correlations. Finally, using neural decoding in electrocorticography, using a combination of in house and publicly available datasets, they find that decoding scores are comparable for sDM as compared to standard methods.

This is an interesting study using a novel method for decomposition/decoding. I have a few questions/comments however.

1. For clarification for the reader (and me), how does dynamic mode decomposition differ from empirical mode decomposition?
2. Fig 1 c: what is the difference between DMD component 1 and DMD component 2? What is meant to be the difference? It isn't clear from the figure/text.
3. Figure 2d: what is the dip in the power spectrum?
4. Figure 2e: How was the across trial analysis done? All pairs? Please clarify.
5. Figure 2 d,e: Why are your correlation values z-scored? Against what?

6. Page 18 Lines 350-358: It would be helpful to show this result in figure form.
7. Figure 4: Are any statistical analyses done on these comparisons? If so, please state, and if not, please state why.
8. What is the computational cost benefit of frequency filtered sDM features? Doesn't this eliminate (some of/all) the computational speed benefits (having to filter)?
9. It would be useful to have a figure/illustration of the computational time benefits of sDMs as compared to standard methods since this is one of the main stated benefits.
10. How were your channels selected for your neural decoding? It is listed in Supp Table 1,2,, but it is not clear how these were chosen.
11. It would be useful to have a direct comparison for the electrode importance between the methods (perhaps do a comparison analysis between the two plots in Supp. Figure 1b).

Reviewer #3 (Remarks to the Author):

The manuscript investigates an interesting topic: using the Koopman modes for ECoG decoding. A new spatial dynamic mode metric is proposed to reduce the computational cost. The following aspect should be considered before it can be accepted to publish.

1. The new sDM features seems very similar to the symmetric positive defined matrices (SPD) in the widely used Riemannian manifold-based methods. A detailed comparison of these two methods should be discussed.
2. Authors claimed that the sDM were derived from the Grassman projection kernels. However, the ψ defined in eqn (5) seems not the same thing of the transformation ψ in eqn (4). This needs to be clarified.
3. For the interpretation in Fig. 2(a), it is hard to see the features are clustered by classes. For example, there seems more correlation between Class 1 and 2, than those of Class 1 and Class 1. This needs to be clarified.
4. From Fig 2(d), the discriminative ability looks similar in a very wide band. It is hard to say, there is a peak value at 80Hz. Additionally, it needs to be clarified why there are undershoots at ~ 160 Hz and 300Hz. P.s. where is Fig 2(c)?
5. The new methods were mainly compared with itself and some very old results in 2016. It should be compared with SOTA methods.

Point-by-Point Replies to Reviewers
Fast, accurate, and interpretable decoding of electrocorticographic
signals using dynamic mode decomposition
(COMMSBIO-23-4569A)

We appreciate the reviewers' helpful comments and suggestions. We also appreciate the constructive feedback and believe it has helped us clarify and improve the manuscript. We have thoroughly revised our manuscript to address all issues raised by the reviewers. In the revised manuscript, added or corrected text appears in **yellow**. Moreover, during the revision, we noticed that the computation time labeled using sDM features with L2-SVM in Fig. 3a was actually the computation time for the case of using snDM and seDM features with L2-SVM. The figure and the corresponding computational complexities (for training: changed from $O(n^{1.11})$ to $O(n^{1.15})$; for testing: changed from $O(n^{0.03})$ to $O(n^{0.05})$) have been corrected in the revised manuscript; however, these changes do not affect the conclusion. In addition, we have also corrected the typing error "reproductivity" on line 289 with "reproducibility". Furthermore, the term "arm movement task" in the title of Fig. 2 has been unified with "arm motor task"; accordingly, we have also removed redundant text in lines 811–812 ("for the ECoG datasets of the arm motor task and hand movement task" in the original text had been changed to "for the ECoG dataset of the arm motor task"). We sincerely apologize for these mistakes. Our point-by-point responses to the comments are listed below, and the reviewer's comments are shown in gray.

Response to Reviewers

Reviewer #1 (Remarks to the Author):

Comment 1

Thank you for inviting me to review this interesting manuscript. The present study describes a novel adaptation of dynamical modes for invasive brain signal decoding. The topic is of general relevance and the methods are clearly described. I only have three major points.

Response to comment 1:

We appreciate Reviewer 1's assessment of our study. We respond to each comment as follows.

Comment 2

1) In addition to the demonstrated comparisons, the authors should compare their results to the best individual channel. This is relevant because for future brain computer interfaces, the number of channels will be very limited and therefore any measure that requires complex and variable spatial locations should clearly outperform a more simple single electrode approach. Additionally, the authors should investigate the performance and stability with relation to number of available channels, e.g. by systematically removing channels. This is relevant to inform the design of brain computer interfaces that could benefit from the approach.

Response to comment 2:

We appreciate this comment, as we agree that the number of available channels directly affects performance. To evaluate the classification accuracy when the number of channels is small, we repeatedly calculated the classification accuracies with randomly selected electrodes. Interestingly, the sDM features showed higher classification accuracy than did the high- γ power features even with a small number of electrodes. This result is now shown in Supplementary Fig. 3b. Moreover, to show how DMD utilizes spatial information from spatiotemporal ECoG signals, we calculated the classification accuracies achieved when varying the number of electrodes applied simultaneously for DMD. It was found that the classification accuracy decreased to near the level of chance when DMD was applied to individual channels; on the other hand, when DMD was applied to 30 individual groups of ECoG signals, each consisting of two channels, the classification accuracy became higher than that achieved using high- γ power features. Hence, the results suggest that DMD successfully utilizes spatial information. This finding is included in Supplementary Fig. 3a. In addition, to reveal how the individual channels contributed to the classification accuracy, we repeatedly performed classification analyses by removing one of the feature components (high- γ power features or snDM features) or the ECoG signals of one channel. For the high- γ power features and snDM features, the contributing electrodes were observed around the sensorimotor cortex, whereas the contributing electrodes were widely distributed for the ECoG signals of individual channels. These findings are shown in Supplementary Fig. 4, which is referenced in the main text along with Supplementary Fig. 3 as follows (lines 360–365):

Therefore, these results demonstrated that the sDM features, especially the snDM features, with the L1-SVM model improved the classification accuracy and computational time for the neural decoding of ECoG signals recorded during hand movements (for the specific effects of the available electrodes on the classification accuracy and the importance of each electrode, see Supplementary Figs. 3 and 4).

Supplementary Fig. 3:

Supplementary Fig. 4:

Comment 3

2) The authors should discuss the opportunities for across patient decoding, which is an increasingly recognized problem with some solutions at hand, e.g. see <https://doi.org/10.1088/1741-2552/abda0b> & <https://doi.org/10.21203/rs.3.rs-3212709/v1>.

Response to comment 3:

We also agree that across-patient decoding has recently become a topic of great interest. In accordance with this comment, we added the following paragraph just before the Conclusions (lines 490–503):

Finally, the proposed sDM features can be further combined with different

machine learning algorithms other than simple SVM and regression. Many new algorithms have been proposed for decoding ECoG signals, including methods based on deep neural networks (DNNs) with long short-term memory²⁶, recurrent neural networks²⁰, and gradient boosting trees²⁷. In fact, it has been reported that the decoding accuracy (correlation coefficient) for the finger flexion task can reach 0.50 for all fingers and for all patients in a dataset with gradient boosting trees²⁷, whereas the accuracy was 0.37 for snDM features with L2-regularized regression (see Supplementary Table 4). Thus, the proposed sDM features may improve the decoding accuracy when combined with these algorithms. Moreover, recent studies using DNNs have shown the viability of across-patient decoding by means of weights for individual patients (subject blocks)²⁸ or electrode-level data projections onto predefined brain regions²⁹. With these techniques, sDM features, which capture the spatiotemporal patterns of multiple signals, could similarly be used for decoding across patients.

Comment 4

3) The authors should make an effort to provide interpretability with respect to source location in addition to the frequency band.

Response to comment 4:

We agree that visualization of the features is important. In the revised manuscript, we have included visualizations of the snDM features and F values for different movement classes on the normalized brain, as shown in Fig. 2d and e, respectively. These panels are referenced in the main text on lines 247–248 as follows:

Moreover, these snDM features and the corresponding F values exhibited high values around the sensorimotor cortex (Fig. 2d, e).

We have also visualized the frequency-filtered snDM features and their corresponding F values in Supplementary Fig. 5, which is referenced on lines 369–373 as follows:

To assess the differences in the frequencies of the modes for different movement types, the classification accuracy was evaluated using snDM features calculated from DMs whose frequencies fell within a given frequency

range (Fig. 1f; for visualization of the frequency-filtered snDM features, see Supplementary Fig. 5).

Fig. 2:

Captions of Figs. 2d and e:

(d, e) The snDM features shown in (b) and their corresponding F values (diagonal part of (c)) are shown by corresponding colors at the location of each electrode on the normalized brain in (d) and (e), respectively.

Supplementary Fig. 5

Reviewer #2 (Remarks to the Author):**Comment 1**

“Fast, accurate, and interpretable decoding of electrocorticographic signals using dynamic mode decomposition” by Fukuma and colleagues seeks to validate a novel method of signal decomposition – dynamic mode decomposition (DM) via spatial DM (sDM) features for neural decoding. They first find that sDMS correlate most highly with higher frequency features ($> \sim 50$ Hz) of the neural signal. They also find that as compared to features in this frequency range, sDMs show more robust trial-wise correlations. Finally, using neural decoding in electrocorticography, using a combination of in house and publicly available datasets, they find that decoding scores are comparable for sDM as compared to standard methods.

This is an interesting study using a novel method for decomposition/decoding. I have a few questions/comments however.

Response to comment 1:

We greatly appreciate Reviewer #2’s interest in our manuscript. We respond to each point raised by Reviewer #2 as follows.

Comment 2

1. For clarification for the reader (and me), how does dynamic mode decomposition differ from empirical mode decomposition?

Response to comment 2:

We appreciate this comment, as this question raised by Reviewer #2 will be of interest to many readers. Although empirical mode decomposition (EMD) is a traditional and well-known method for signal decomposition, EMD is difficult to apply to high-dimensional data due to its computational cost. In addition, the decompositions produced by EMD are difficult to interpret because they take the form of a specific function due to the Hilbert–Huang transformation. In contrast, the decompositions produced by DMD are known to be closely related to several physical concepts. To clarify this point, we have added the following paragraph to the discussion (lines 466–488):

The proposed method utilizes information about the nonlinear dynamics in ECoG signals while avoiding the computational costs incurred for decoding via kernel methods. In our previous study, decoding was performed based on a symmetric positive definite matrix (Gram matrix) calculated by means of a projection kernel on a Riemannian manifold; in contrast, the proposed method directly converts the DMs into sDM features, enabling the use of non-kernel-based decoders (e.g., L1-regularized SVM). It should also be mentioned that both DMD and traditional empirical mode decomposition (EMD) offer effective ways to derive information about the nonlinear dynamics in time-series data. However, DMD has the following main advantages over EMD for high-dimensional data with nonlinearities, such as in the present case. (1) The decompositions obtained by DMD are known to be closely related to several physical concepts, such as phase reduction as discussed in the field of nonlinear science, allowing the extracted information to be interpreted both physically and mathematically. (2) EMD is a method that captures nonlinearity and nonstationarity in data based on the extraction and smoothing of local extreme values in the data, and in principle, it is known to be numerically unstable and generally difficult to apply to high-dimensional data due to its large computational cost. On the other hand, DMD is based on relatively stable numerical computations (SVD and eigenvalue decomposition), is known to be robust for high-dimensional data, and is capable of extracting information that captures global dynamic features in data⁶. In fact, the proposed sDM features are informative for various types of tasks and decoding (Fig. 4). Considering the improved computational cost in this study, the proposed method may also be effective even for different modalities of spatiotemporal signals with high spatial dimensions.

Comment 3

2. Fig 1 c: what is the difference between DMD component 1 and DMD component 2? What is meant to be the difference? It isn't clear from the figure/text.

Response to comment 3:

We apologize for the unclear explanation in the figure caption. DMD component 1 and DMD component 2 are complex conjugate pairs, possessing the same real values but different imaginary parts whose signs are inverted. Because the original spatiotemporal

signal $\mathbf{x}(t)$ takes real values, the DMD components always include complex conjugate pairs. To clarify these points, we have revised the caption of Fig. 1c and d as follows:

(c, d) Four DMD components determined based on the four SVD components with nonzero singular values are shown for (c) time dynamics and DMs, along with (d) their corresponding spatiotemporal signals. Here, DMD components 1 and 2 and DMD components 3 and 4 are complex conjugate pairs with respect to their modes and temporal dynamics because the original spatiotemporal signal $\mathbf{x}(t)$ is strictly real. For visibility, each DM was L2-normalized, and the scaling factor for each DM was applied to the corresponding time dynamics so that their products were the same. By adding the products of the four DMs and time dynamics in (c), the original signals ($\mathbf{x}(t)$) were reconstructed ($\mathbf{x}_{\text{recon}}(t)$) as shown in (d).

Comment 4

3. Figure 2d: what is the dip in the power spectrum?

Response to comment 4:

We appreciate this comment, as the dips in the correlation coefficient between the snDM features and the power spectrum density are obvious but lack explanation. Dips exist at 180 Hz and 300 Hz, which are odd harmonics of the power supply frequency (60 Hz). Hence, these observations suggest that power line noise reduced the consistency of the measured ECoG signals at these frequencies, resulting in lower correlation coefficients. We have added the following sentence to the caption of Fig. 2f, which was Fig. 2d in the original manuscript.

The dips in correlation at 180 Hz and 300 Hz are considered to be due to noise caused by odd harmonics of the power supply frequency (60 Hz).

Comment 5

4. Figure 2e: How was the across trial analysis done? All pairs? Please clarify.

Response to comment 5:

We apologize for the unclear explanation of the analysis. The reproducibility was calculated for all possible pairs of the same movement for each patient. To clarify this point, we have revised the main text on lines 276–279 and 284–290 as follows:

Because each component of the snDM features corresponds to a channel, the snDM features of all trials were concatenated **for each patient** to calculate Pearson's correlation coefficient, with the PSD of each frequency concatenated for all channels and trials.

On the other hand, when the snDM features were compared **against** different trials of the same movement, **considering all possible pairs for each patient**, the reproducibility of the snDM features was significantly higher than that of the power features, including the high- γ power features (Fig. **2g**; $p < 1.0 \cdot 10^{-21}$, $F(4,50)=94.95$, one-way ANOVA; snDM features vs. other features, $p < 6.4 \cdot 10^{-8}$, post hoc Tukey–Kramer test; for **reproducibility** during visual perception, see Supplementary Fig. 1).

Comment 6

5. Figure 2 d,e: Why are your correlation values z-scored? Against what?

Response to comment 6:

We sincerely apologize for the lack of information here. The correlation coefficients were Fisher z-transformed before averaging to avoid a highly skewed distribution of the Pearson correlation coefficients. To clarify this point, we have updated the caption of Fig. 2f, which corresponds to Fig. 2d in the original manuscript, as follows:

The correlation coefficients were Fisher z-transformed and averaged among the patients, as shown by the black line, with the colored area representing the 95% confidence intervals (CIs) among the patients.

Moreover, we have updated the caption of Fig. 2g, which was Fig. 2e in the original manuscript, as follows:

To calculate the reproducibility of each feature, the correlation coefficients were calculated among all possible pairs of trials of the same movement type, Fisher z -transformed, and averaged for each patient.

Comment 7

6. Page 18 Lines 350-358: It would be helpful to show this result in figure form.

Response to comment 7:

We agree that it would be helpful to readers to show the significant differences in figure form. In accordance with this comment, we have added significant difference markers in Fig. 3c as follows:

Fig. 3:

We have also updated the caption of Fig. 3c as follows:

(c) Frequency-filtered snDM features were calculated for each frequency band of the ECoG signals to perform classification with the L1-SVM model with an optimized rank. Classification was also performed based on the features created by concatenating all of the frequency-filtered snDM features (combined) and snDM features without frequency filtering (nonfiltered). The average classification accuracies are shown by bars with 95% CIs among the patients. The differences in the classification accuracies among the frequency bands were evaluated by one-way ANOVA with a post hoc Tukey–Kramer test. $*p < 0.05$, $**p < 0.01$.

Comment 8

7. Figure 4: Are any statistical analyses done on these comparisons? If so, please state, and if not, please state why.

Response to comment 8:

We appreciate this comment. In this figure, we intended to show that the snDM features perform better than or similar to the high- γ power features for different tasks and different types of decoding. In response to comment 6 from Reviewer #3, we have modified Fig. 4 so that each panel shows the decoding accuracy achieved using the high- γ power features instead of the reported accuracy. In the updated figure, we have also added the results of statistical tests between the decoding accuracy achieved using the high- γ power features and that achieved using the snDM features, in response to this comment.

We have also revised the caption of Fig. 4 in accordance with the changes in the figure as follows:

(a-d) Bars show decoding accuracies using **high- γ power features**, snDM features, seDM features, and the combination of both features (black bars) and those using the corresponding part of the frequency-filtered sDM features (white bars) for **the** ECoG datasets: (a) hand versus tongue movement task (Miller, 2019), (b) image perception task (Miller, 2019), (c) finger flexion task (Miller, 2019; **the average of the Fisher z-transformed correlation coefficients for the thumb, index finger, and little finger is shown, following** Miller et al., 2009¹⁹; **for the**

correlation coefficients for each patient and each finger, see Supplementary Table 4), and (d) video perception task (Fukuma et al., 2022; the average of the Fisher z -transformed correlation coefficients among 1,000 dimensions is shown). For classification in (a) and (b), L1-SVM was applied. The differences in the decoding accuracies using the high- γ power features and snDM features were evaluated by means of two-tailed paired t tests. $*p < 0.05$, $**p < 0.01$.

Accordingly, the main text has been modified as follows (lines 392–394, lines 398–402, and lines 875–876, respectively):

The accuracies of neural decoding using snDM and seDM features based on ECoG signals were compared among different types of tasks and with the accuracies using high- γ power features.

As summarized in Fig. 4, the proposed method successfully decoded ECoG signals with an accuracy higher than or comparable to that using high- γ power features regardless of task or decoding type (due to the computational time, L2 regularization was used for regression throughout this study; Supplementary Fig. 6 and Supplementary Tables 3–4).

The decoding accuracies using the high- γ power features and the snDM features were compared using a two-tailed paired t test (Fig. 4).

Comment 9

8. What is the computational cost benefit of frequency filtered sDM features? Doesn't this eliminate (some of/all) the computational speed benefits (having to filter)?

Response to comment 9:

We sincerely apologize for the unclear explanation of the filtering process. Here, the term “frequency filtering” refers to the selection of DMs based on their frequencies, not to filtering of the original ECoG signals with signal processing filters such as bandpass or low-pass filters. Because only the DM selection process is additionally performed for the acquisition of the frequency-filtered sDM features compared to the process of obtaining the original sDM features, the added computational cost is considered

negligible. To clarify the meaning of the term “frequency filtering” here, we have revised the main text as follows (lines 369–374):

To assess the differences in the frequencies of the modes for different movement types, the classification accuracy was evaluated using snDM features calculated from DMs whose frequencies fell within a given frequency range (Fig. 1f; for visualization of the frequency-filtered snDM features, see Supplementary Fig. 5). Here, the evaluation was performed with the L1-SVM model for conventional frequency bands (0–1, 1–4, 4–8, 8–13, 13–30, 30–80, 80–150, and 150–500 Hz).

Comment 10

9. It would be useful to have a figure/illustration of the computational time benefits of sDMs as compared to standard methods since this is one of the main stated benefits.

Response to comment 10:

We agree that a direct comparison of the computation times for sDMs and standard methods will be of interest to readers. In the revised manuscript, we have added Supplementary Fig. 2 showing the training and testing times for decoders using sDM and high- γ power features, and this figure is now referenced on lines 311–315 in the main text as follows:

Similarly, the prediction time for a new sample increased with the number of training samples ($\sim O(n^{0.75})$) for the Gram matrix, while the prediction time using the sDM features and linear L2-SVM was much shorter and increased much more slowly with the number of training samples ($\sim O(n^{0.05})$) (for a comparison of the computational times between high- γ power features and sDM features, see Supplementary Fig. 2).

Supplementary Fig. 2:

In addition to the above modifications, we noticed that the method for evaluating the computational time was missing from the method. The details of the evaluation have been described in the methods of the main text as follows (lines 851–861):

Evaluation of computational time

Decoder training time and decoder testing time on a new sample were assessed by varying the number of samples per class (n) using the precomputed DMs of patient 1 in the arm motor task. First, trials were randomly resampled so that the number of trials in each class was equal to n , and then the DMs of the resampled trials were used to train the decoder model while training time was measured. The cost parameter for training was selected as the most frequent value among the optimized costs in the outer folds of the nested cross-validation to calculate the classification accuracy. To measure the testing time of the decoder, the precomputed DMs of a randomly selected trial were applied to the decoder. The measurement was repeated 100 times by changing the seed value for the random number generator.

Comment 11

10. How were your channels selected for your neural decoding? It is listed in Supp Table 1,2,, but it is not clear how these were chosen.

Response to comment 11:

We appreciate this important comment. For this study, we used our in-house dataset of ECoG signals, which includes data from patients who had electrodes implanted in regions including the sensorimotor cortex. The electrode placement was solely determined by clinical necessity. The dataset was created in our previous study

(Shiraishi et al., 2020) by selecting the ECoG signals of the frontal and parietal cortices. To clarify this point, we have revised the main text as follows (lines 213–218):

This dataset is composed of ECoG signals that were recorded at 1 kHz while 11 patients performed three types of movements with their hand contralateral to the implanted electrodes. Due to clinical requirements, all these patients had subdural electrodes implanted in cortical areas, including the sensorimotor cortex. The dataset consists of ECoG signals from the frontal and parietal cortices (ECoG dataset of arm motor task; Supplementary Table 1).

In addition, we have revised the methods as follows (lines 527–529 and lines 553–557):

The ECoG dataset of the arm motor task consisted of eleven subjects (7 males; age range, 13–66 years), with subdural electrodes placed on their front-parietal area, including the sensorimotor cortex.

Signal preprocessing of this dataset was performed as described in Shiraishi et al. (2020) by rejecting noisy channels and channels located outside of the front-parietal area via visual inspection (for the number of channels in the dataset, see Supplementary Table 1). For the analyses in this study, the ECoG signals were common-average referenced and cropped from 0 to 500 ms with respect to the movement cue.

To clarify that the same electrodes used in our previous study (Fukuma et al., 2022) were used in this manuscript, we also revised the section describing the signal preprocessing of the ECoG dataset for the video perception task as follows (lines 635–640):

Signal preprocessing of this dataset was performed as described in Fukuma et al. (2022) by rejecting noisy channels via visual inspection (for the number of channels in the dataset, see Supplementary Table 2). The ECoG signals in the dataset were filtered with a lowpass filter (8th-order Chebyshev Type I infinite impulse response filter) and downsampled to 1 kHz. The downsampled ECoG signals were then rereferenced by common averaging.

Comment 12

11. It would be useful to have a direct comparison for the electrode importance between the methods (perhaps do a comparison analysis between the two plots in Supp. Figure 1b).

Response to comment 12:

We sincerely agree that the electrode importance will be of interest to readers. To respond to this comment, we repeatedly performed decoding analyses by removing one of the feature components (high- γ power (80–150 Hz) features and snDM features) or removing ECoG signals from one electrode (snDM features with no information from one electrode) to determine the electrode importance. The results are now shown in Supplementary Fig. 4, which is referenced in the main text as follows (lines 360–365):

Therefore, these results demonstrated that the sDM features, especially the snDM features, with the L1-SVM model improved the classification accuracy and computational time for the neural decoding of ECoG signals recorded during hand movements (for the specific effects of the available electrodes on the classification accuracy and the importance of each electrode, see Supplementary Figs. 3 and 4).

Supplementary Fig. 4:

Reviewer #3 (Remarks to the Author):

Comment 1

The manuscript investigates an interesting topic: using the Koopman modes for ECoG decoding. A new spatial dynamic mode metric is proposed to reduce the computational cost. The following aspect should be considered before it can be accepted to publish.

Response to comment 1:

We greatly appreciate Reviewer #3's interest in our manuscript. We respond to each point raised by Reviewer #3 as follows.

Comment 2

1. The new sDM features seems very similar to the symmetric positive defined matrices (SPD) in the widely used Riemannian manifold-based methods. A detailed comparison of these two methods should be discussed.

Response to comment 2:

We appreciate this comment, as clarifying this point will surely make the manuscript easier to understand. In fact, the Gram matrix in Fig. 2a is a symmetric positive definite (SPD) matrix acquired through a Riemannian manifold-based method (the projection kernel method). Hence, our previous study (Shiraishi et al., 2020) was based on the SPD approach. In contrast, in this study, we attempted to improve the computational speed and interpretability by means of mathematical operations corresponding to the projection kernel method and L2-SVM. To clarify this point, we have added the following paragraph to the discussion (lines 466–488).

The proposed method utilizes information about the nonlinear dynamics in ECoG signals while avoiding the computational costs incurred for decoding via kernel methods. In our previous study, decoding was performed based on a symmetric positive definite matrix (Gram matrix) calculated by means of a projection kernel on a Riemannian manifold; in contrast, the proposed method directly converts the DMs into sDM features, enabling the use of non-kernel-based decoders (e.g., L1-regularized SVM). It should also be mentioned that both DMD and traditional empirical mode decomposition (EMD) offer effective ways to derive information about the nonlinear dynamics in time-series data. However, DMD has the following main advantages over EMD for high-dimensional data with nonlinearities, such as in the present case. (1) The decompositions obtained by DMD are known to be closely related to several physical concepts, such as phase reduction as discussed in the field of nonlinear science, allowing the extracted information to be interpreted both physically and mathematically. (2) EMD is a method that captures nonlinearity and nonstationarity in data based on the extraction and smoothing of local extreme values in the data, and in principle, it is known to be numerically unstable and

generally difficult to apply to high-dimensional data due to its large computational cost. On the other hand, DMD is based on relatively stable numerical computations (SVD and eigenvalue decomposition), is known to be robust for high-dimensional data, and is capable of extracting information that captures global dynamic features in data⁶. In fact, the proposed sDM features are informative for various types of tasks and decoding (Fig. 4). Considering the improved computational cost in this study, the proposed method may also be effective even for different modalities of spatiotemporal signals with high spatial dimensions.

Comment 3

2. Authors claimed that the sDM were derived from the Grassman projection kernels. However, the Ψ defined in eqn (5) seems not the same thing of the transformation Ψ in eqn (4). This needs to be clarified.

Response to comment 3:

We sincerely apologize for the use of the Φ notation, which might not be intelligible. Through the use of Φ^i , we intended to describe all the DMs (a matrix created by concatenating all the DMs) for the i -th trial and not only part of the DMs. Hence,

$\psi(\Phi^i)^\dagger \psi(\Phi^j)$ in the last line of Equation (4) can be rewritten using Equation (5)

($\psi(\Phi) = \text{vec}(\Phi\Phi^\dagger)$) as follows:

$$\psi(\Phi^i)^\dagger \psi(\Phi^j) = \left(\psi(\Phi^i)\right)^\dagger \left(\psi(\Phi^j)\right) = \left(\text{vec}((\Phi^i)(\Phi^i)^\dagger)\right)^\dagger \left(\text{vec}((\Phi^j)(\Phi^j)^\dagger)\right),$$

which can be simplified to $\text{vec}(\Phi^i\Phi^{i\dagger})^\dagger \text{vec}(\Phi^j\Phi^{j\dagger})$ in the second line of Equation (4).

To clarify that the superscript of Φ denotes a particular trial, we have revised the main text as follows (lines 170–172):

The projection kernel k_p between the two matrices of DMs for the i -th and j -th trials (Φ^i and Φ^j , respectively; $\Phi^i, \Phi^j \in \mathbb{C}^{P \times K}$) is written as follows with \mathbf{A}^\dagger denoting conjugate transpose of \mathbf{A} .

Comment 4

3. For the interpretation in Fig. 2(a), it is hard to see the features are clustered by classes. For example, there seems more correlation between Class 1 and 2, than those of Class 1 and Class 1. This needs to be clarified.

Response to comment 4:

We sincerely apologize for the misleading use of the term “clustering”. What we meant here is that the kernel values against other trials became similar among the trials for the same movement type. In addition, because the Gram matrix in Fig. 2a is composed of the kernel values of the projection kernel, the values among the same movement class are not always high. To clarify this point, we have revised the main text on lines 233–234 as follows:

The Gram matrix showed that the kernel values became similar among trials for the same movement type.

Comment 5

4. From Fig 2(d), the discriminative ability looks similar in a very wide band. It is hard to say, there is a peak value at 80Hz. Additionally, it needs to be clarified why there are undershoots at ~160Hz and 300Hz. P.s. where is Fig 2(c)?

Response to comment 5:

We agree that the use of the word “peak” is inappropriate for describing the high values between approximately 80 Hz and 200 Hz. To clarify our intended meaning, we have revised the main text as follows (lines 279–282):

The correlation coefficients were Fisher z-transformed and averaged among all patients; the resulting coefficients became high from ~80 Hz to ~200 Hz (Fig. 2f), a range that interestingly includes the high- γ band (80–150 Hz), which is known to be the most informative frequency band for movement classification¹⁸.

Additionally, the frequencies of the undershoots in Fig. 2f (Fig. 2d in the original manuscript) are 180 Hz and 300 Hz, which are odd harmonics of the power supply frequency (60 Hz). Hence, it is considered that noise due to these odd harmonics caused

deterioration in the ECoG signals at these frequencies, resulting in inconsistent measured brain activity, which further led to lower correlation coefficients between the power and snDM features. In the revised manuscript, we mention the power supply frequency in the caption of Fig. 2f as follows:

The dips in correlation at 180 Hz and 300 Hz are considered to be due to noise caused by odd harmonics of the power supply frequency (60 Hz).

Finally, we sincerely apologize for the panel layout of Fig. 2. In the revised manuscript, we have added more space around Panels b and c so that they can be identified easily.

Fig. 2:

Comment 6

5. The new methods were mainly compared with itself and some very old results in 2016. It should be compared with SOTA methods.

Response to comment 6:

We appreciate this comment. What we intended to show throughout this paper (especially in Figs. 3 and 4) is that our proposed snDM features perform better than or similar to the high- γ power features for different tasks and different types of decoding. To clarify this point, we have replaced the “reported” accuracy in each panel of Fig. 4 with the decoding accuracy achieved using the high- γ power features. In accordance with this change, Fig. 4, its caption, and the main text have been revised as follows:

Caption of Fig. 4:

(a-d) Bars show decoding accuracies using high- γ power features, snDM features, seDM features, and the combination of both features (black bars) and those using the corresponding part of the frequency-filtered sDM features (white bars) for the ECoG datasets: (a) hand versus tongue movement task (Miller, 2019), (b) image perception task (Miller, 2019), (c) finger flexion task (Miller, 2019; the average of the Fisher z-transformed correlation coefficients for the thumb, index finger, and little finger is shown, following Miller et al., 2009¹⁹; for the correlation coefficients for each patient and each finger, see Supplementary Table 4), and (d) video perception task (Fukuma et al., 2022; the average of the Fisher z-transformed correlation coefficients among 1,000 dimensions is shown). For classification in (a) and (b), L1-SVM was applied. The differences in the decoding accuracies using the high- γ power features and snDM features were evaluated by means of two-tailed paired t tests. * $p < 0.05$, ** $p < 0.01$.

Main text (lines 392–394, lines 398–402, and lines 875–876):

The accuracies of neural decoding using sNDM and seDM features based on ECoG signals were compared among different types of tasks and with the accuracies using high- γ power features.

As summarized in Fig. 4, the proposed method successfully decoded ECoG signals with an accuracy higher than or comparable to that using high- γ power features regardless of task or decoding type (due to the computational time, L2 regularization was used for regression throughout this study; Supplementary Fig. 6 and Supplementary Tables 3–4).

The decoding accuracies using the high- γ power features and the sNDM features were compared using a two-tailed paired t test (Fig. 4).

Despite these changes, we still understand that referencing SOTA methods is important; hence, we refer to SOTA methods and their accuracy in a newly added paragraph in the discussion as follows (lines 490–503):

Finally, the proposed sDM features can be further combined with different machine learning algorithms other than simple SVM and regression. Many new algorithms have been proposed for decoding ECoG signals, including methods based on deep neural networks (DNNs) with long short-term memory²⁶, recurrent neural networks²⁰, and gradient boosting trees²⁷. In fact, it has been reported that the decoding accuracy (correlation coefficient) for the finger flexion task can reach 0.50 for all fingers and for all patients in a dataset with gradient boosting trees²⁷, whereas the accuracy was 0.37 for sNDM features with L2-regularized regression (see Supplementary Table 4). Thus, the proposed sDM features may improve the decoding accuracy when combined with these algorithms. Moreover, recent studies using DNNs have shown the viability of across-patient decoding by means of weights for individual patients (subject blocks)²⁸ or electrode-level data projections onto predefined brain regions²⁹. With these techniques, sDM features, which capture the spatiotemporal patterns of multiple signals, could similarly be used for decoding across patients.

REVIEWERS' COMMENTS:

Reviewer #1 (Remarks to the Author):

The authors have thoroughly addressed my comments.

Reviewer #2 (Remarks to the Author):

I thank the authors for their thoughtful work in addressing my comments/concerns. They have all now been addressed.

Reviewer #3 (Remarks to the Author):

The authors have addressed all my concerns. I have no further comments.